# ALTo: Adaptive-Length Tokenizer for Autoregressive Mask Generation

**Lingfeng Wang**[*,1,2,◇], **Hualing Lin**[*,2], **Senda Chen**[*,3], **Tao Wang**[*,1], **Changxu Cheng**[1,†],
**Yangyang Zhong**[2], **Dong Zheng**[1,2], **Wuyue Zhao**[1,†]

[1]Uni-Ubi    [2]Zhejiang University    [3]Tongji University

[*]Equal contributions    [†]Corresponding author    [◇]Work done during internship at Uni-Ubi

{yayafengzi, linhualing, zhongyangyang, ddzheng}@zju.edu.cn,
{sendachen586, ccx0127}@gmail.com, {wangtaomarvel, zhaohongyi}@uniubi.com

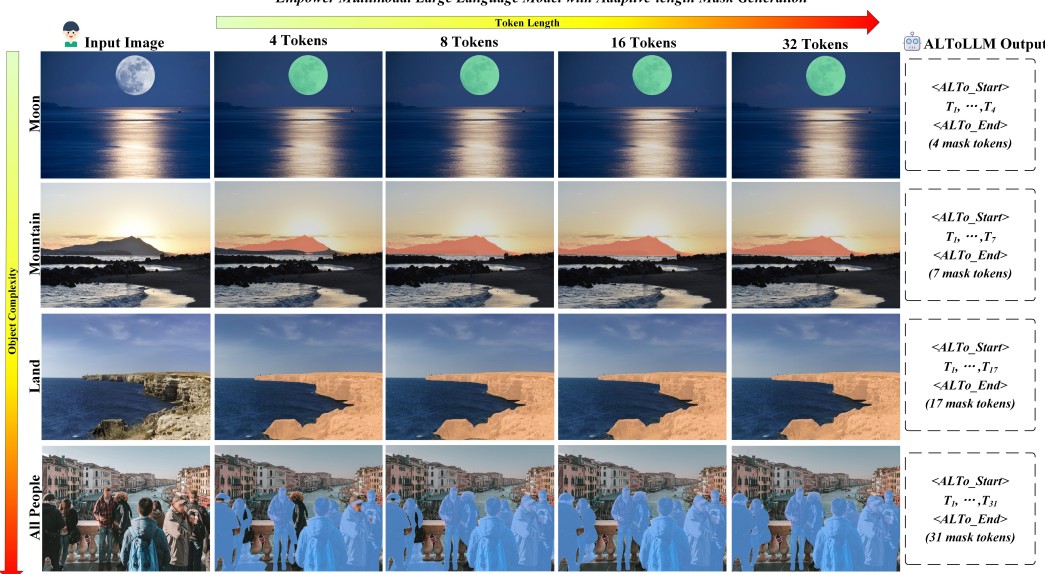

Figure 1: ALToLLM realizes adaptive-length mask token generation according to object complexity.

## Abstract

While humans effortlessly draw visual objects and shapes by adaptively allocating attention based on their complexity, existing multimodal large language models (MLLMs) remain constrained by rigid token representations. Bridging this gap, we propose ALTo, an adaptive-length tokenizer for autoregressive mask generation. To achieve this, a novel token length predictor is designed, along with a length regularization term and a differentiable token chunking strategy. We further build ALToLLM that seamlessly integrates ALTo into MLLM. Preferences on the trade-offs between mask quality and efficiency is implemented by group relative policy optimization (GRPO). Experiments demonstrate that ALToLLM achieves state-of-the-art performance with adaptive token cost on popular segmentation benchmarks. Code and models are released at https://github.com/yayafengzi/ALToLLM.

## 1   Introduction

Multimodal large language models (MLLMs) have demonstrated remarkable capabilities in image and text understanding tasks. However, their generative abilities remain largely limited to text [1, 2, 3, 4, 5, 6]. Given the inherent differences between text and image modalities, introducing

39th Conference on Neural Information Processing Systems (NeurIPS 2025).

a lightweight image decoder into multimodal understanding models to enable image generation remains a significant challenge. To align with the next-token prediction paradigm of text generation, discretizing images using image tokenizers (e.g., VQGAN [7]) has become a natural and effective approach. Within this framework, various visual modalities—including RGB images, segmentation masks, and depth maps—can be uniformly represented as "images", enabling unified modeling for both multimodal understanding and generation [8, 9].

Early image tokenizers typically represent images using fixed-length token sequences without considering the inherent complexity of the images [7, 10, 11, 12]. This fixed-length design may lead to insufficient representation for complex images while generating redundant tokens for simpler ones, resulting in resource wastage and reduced efficiency. In contrast, humans can flexibly allocate attention based on the complexity of the task [13]. For example, segmenting complex shapes requires more attention and effort compared to simpler shapes.

Recent arts are dedicated to learning hierarchical and flexible tokens [14, 15, 16]. The representations become increasingly fine-grained as the number of tokens increases. Based on our observations, the number of tokens required to represent fine-grained edge shapes can vary drastically depending on their complexity.

In recent years, several studies [19, 20] have explored adaptive-length tokenization for image representations. The problem, however, is that they all determine adaptive lengths by relying on heuristic rules conditioned on the input image, rather than allowing the model to decide on its own. Although this is feasible in image tokenization, it becomes impractical for image generation since the reconstruction loss is unavailable. As a result, it becomes imperative to enable the model to autonomously determine the adaptive token length, specifically for MLLM scenarios that are expensive in computation, like MLLM-based object segmentation.

Table 1: The flexibility and autonomous adaptivity of different token representation methods. Flexibility refers to hierarchical coarse-to-fine token representation, while autonomous adaptivity denotes spontaneous allocation of token numbers based on object complexity.

| Token Representation Adaptivity | Flexibility | Autonomous |
|---|---|---|
| VAE [17] | × | × |
| VQVAE [10] | × | × |
| VQGAN [7] | × | × |
| TiTok [11] | × | × |
| FlexTok [14] | ✓ | × |
| Emu3 [15] | ✓ | × |
| Chameleon [18] | × | × |
| HiMTok [16] | ✓ | × |
| ALIT [19] | × | × |
| ElasticTok [20] | ✓ | × |
| ALToLLM (ours) | ✓ | ✓ |

To enable adaptive-length modeling for the specific task of mask image generation, we propose ALTo, an **A**daptive-**L**ength **To**kenizer designed for autoregressive mask generation. We further develop ALToLLM. As shown in Fig. 1, ALToLLM is a multimodal large language model (MLLM) that realizes instruction-based mask generation using adaptive-length mask tokens according to object complexity. At the core of ALTo is a novel token length predictor (TLP) embedded within an encoder–VQ–decoder architecture. Given an input mask image, the ALTo encoder is responsible not only for generating discrete tokens but also for predicting the appropriate token sequence length via TLP. To support adaptive-length learning, we introduce a length regularization term and a differentiable token chunking strategy. Together, these enable ALTo to effectively encode masks into variable-length token sequences. To evaluate the effectiveness of ALTo, we construct ALToLLM without any bells and whistles, making no modifications to the underlying LLM architecture or training paradigm. The model is trained using supervised fine-tuning and group relative policy optimization (GRPO) on referring image segmentation tasks. ALToLLM learns to adaptively insert an end-of-mask token (<ALTo_End>) once sufficient mask tokens have been generated. Moreover, GRPO allows dynamic control over token length to balance mask quality and computational efficiency.

In summary, the contributions are as follows:

- We propose ALTo, an adaptive-length mask tokenizer that, for the first time, enables the model to autonomously determine the number of mask tokens based on the complexity of the input mask.

- We develop ALToLLM, which integrates ALTo into a multimodal large language model (MLLM), enabling adaptive mask token generation for object segmentation tasks. The number of generated tokens can vary from as few as 2 to as many as 32, with most cases around 17, allowing ALToLLM to balance quality and efficiency under different scenarios via GRPO.

- Extensive experiments demonstrate that ALTo enables effective and efficient mask image reconstruction, while ALToLLM achieves state-of-the-art performance with adaptive token usage

across various object segmentation benchmarks, including referring expression segmentation and open-vocabulary segmentation.

## 2 Related work

**Visual tokenizers** play important roles in various visual tasks, such as image reconstruction [10, 7], visual compression [20], and visual generation [21, 22, 8, 16]. VQVAE [10, 23] and VQGAN[7] are popular frameworks that encode images into discrete 2D tokens by vector quantization. BEiT [24] exploits visual tokens in masked image modeling. To reduce the redundancy in 2D space, TiTok [11] and SEED [12] produce 1D sequence for image tokenization. Methods above typically use a rigid number of tokens to represent images, regardless of the complexity of the visual content. To have *flexible* tokenization, FlexTok [14] projects images into 1D variable-length token sequences. ElasticTok [20] proposed an adaptive tokenizer for images and videos by dropping a random number of the latter tokens during training. ALIT [19] discretizes the images into flexible-length tokens by recurrent distillation until the reconstruction quality is good or the maximum iterations are met. HiMTok [16] learns 1D hierarchical mask tokens to represent coarse to fine segmentation masks. However, these methods cannot decide an *adaptive* number of tokens autonomously. Trials have been made by heuristic rules about image reconstruction quality [19, 20], which increases computational overhead and becomes impossible for image generation tasks. Our proposed ALTo is both flexible and adaptive, as illustrated in Table 1.

**MLLM-based image segmentation methods** primarily follow three paradigms [25, 16, 26]. MLLM-segmentation joint models such as LISA [27], GSVA [28], GLaMM [29], PixelLM [30], and PSALM [31], create semantic-to-pixel connections through LLM hidden states and rely on additional segmentation modules. Text-based methods, including Text4Seg [25], LLaFS [32] and VistaLLM [33], represent masks as text sequences (pixel classes or polygon vertices), suffering from heuristic and inaccurate mask representation. Interestingly, segmentation masks could also be viewed as images so that we can rethink image segmentation as a mask generation task [8, 9, 34]. HiMTok [16] applies the idea by utilizing a hierarchical mask tokenizer into LLMs. Going a step further, ALToLLM generates adaptive-length token sequences, which is efficient and effective.

**Reinforcement learning (RL)** has become increasingly important for enhancing vision-language models [35, 36, 37, 38, 39]. Approaches like direct preference optimization [40] and proximal policy optimization [41] face challenges with data efficiency and reward stability. Group relative policy optimization (GRPO) [42] has emerged as a promising alternative through its groupwise reward mechanism. Recent applications demonstrate GRPO's effectiveness across various vision-language tasks. Visual-RFT [43] combines GRPO with verifiable rewards for efficient model adaptation. Vision-R1 [44] employs GRPO with progressive thinking suppression for complex reasoning. Seg-Zero [45] achieves zero-shot segmentation through pure RL. These works apply RL to text output, while we make it for preference optimization on mask token output.

## 3 Methods

### 3.1 Overview

The proposed adaptive-length tokenizer (ALTo) represents object masks as token sequences whose lengths adapt to the complexity of objects autonomously. Simple objects (e.g., a sphere) may require few tokens, while intricate structures (e.g., complicated shapes and multiple objects) may use up to 32 tokens. Built on this, ALToLLM is introduced to perform instruction mask generation for referring image segmentation, as shown in Fig. 2. We design a multi-stage training recipe for ALTo and ALToLLM to learn flexible, adaptive, and effective mask representations and achieve strong segmentation performance, as shown in Fig. 3.

**Inference**. As illustrated in Fig. 2, ALToLLM takes as input the image and text by the popular ViT-projector-LLM architecture [1, 46], then autoregressively generates both text tokens and compact mask tokens of adaptive length. The mask tokens along with the pixel-encoded features are fed into the mask de-tokenizer to generate the final mask.

**Training**. As shown in Fig. 3, the training recipe consists of three progressive stages. **Stage 1**: We pretrain the mask tokenizer (MT) and mask de-tokenizer (MD) to reconstruct complex masks

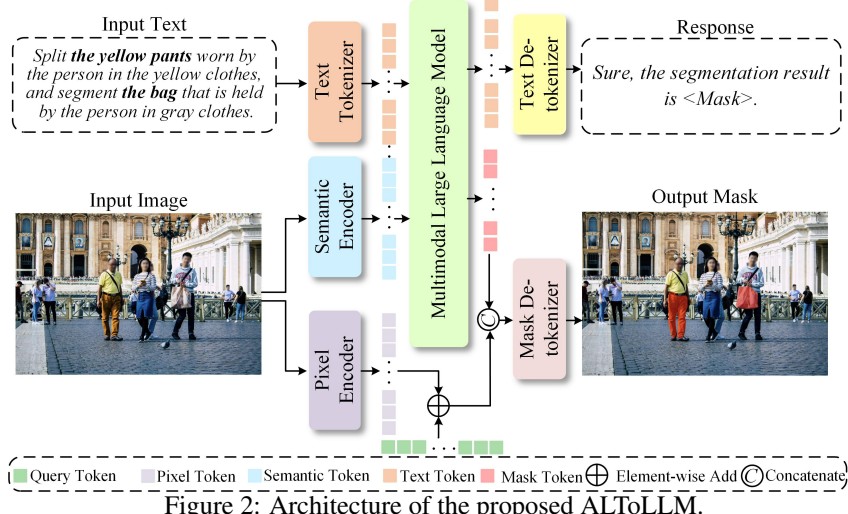

Figure 2: Architecture of the proposed ALToLLM.

using variable-length tokens. **Stage 1.5**: We fine-tune the token length predictor (TLP) to enable adaptive tokenization. **Stage 2**: We leverage ALTo to generate both fixed-length and adaptive-length token labels, which are used to supervise ALToLLM. This equips the model with the basic ability to understand and generate both text and mask tokens. **Stage 3**: We employ GRPO [42] to further adjust specific preferences on trade-off between mask quality and token efficiency. We will introduce the details in the following subsections.

### 3.2 ALTo

The adaptive-length tokenizer (ALTo) comprises three components: a mask tokenizer (MT), a mask de-tokenizer (MD) with a pixel encoder, and a token length predictor (TLP), as shown in Fig. 3 (a). Following HiMTok [16], MT utilizes a transformer encoder with 32 learnable latent tokens to extract information from the input mask and then discretized into 32 mask tokens via vector quantizer (VQ). To support variable-length tokenization, a random number of tail tokens are dropped during training, retaining only the leading tokens. MD is a bidirectional transformer. Differently from HiMTok, MD takes as input the mask tokens and 256 pixel-encoded image features rather than learnable latent tokens. This provides fine-grained guidance for mask generation, inspired by UViM [47]. In training stage 1, the reconstruction is supervised by a mean squared error (MSE) loss $\mathcal{L}_{\text{Mask}}$ to pretrain MT and MD.

The novel TLP determines the optimal number of tokens for each mask. TLP leverages the CLS token feature $T_{\text{cls}}$, which encodes global image features, together with the 32 mask token features $T \in \mathbb{R}^{32 \times d}$, to predict a proper token length. $T_{\text{cls}}$ is used as a query in an attention mechanism to evaluate the importance of each mask token. For each mask token $T_i$, a gated key is generated by SwiGLU as $k_i = (W_{\text{v}}T_i) \odot \sigma(W_{\text{g}}T_i)$. The probability for each token to be the stopping point is computed via scaled dot-product attention: $p = \text{softmax}(q_{\text{cls}}k^T/\sqrt{d})$. The predicted length is computed as the mathematical expectation $\hat{L} = \sum_{i=1}^{32} i \cdot p_i$.

Accordingly, the first $\hat{L}$ tokens are selected and sent to the MD, while the remaining tokens are zero-padded, represented as $H \odot T$, where $H$ is a binary mask defined as $H = \mathbb{I}[i \leq \hat{L}]$. However, such token chunking strategy is not differentiable, which prevents gradients from the mask de-tokenizer from flowing back to the TLP. To address this, we introduce a **differentiable token chunking** strategy by considering the stopping probability distribution $p$. The probability that the $i$-th token is used is given by the cumulative probability $P_i = 1 - \sum_{j<i} p_j$, which indicates that the stop position is later than this token and provides a soft version of token chunking. This inspires us to apply a straight-through estimator as $\hat{T} = (P - P.\text{detach}() + H) \odot T$. In this formulation, the predicted mask is then given by $M_{\text{pred}} = \text{MD}(\hat{T}, X_{\text{img}})$.

In stage 1.5, we use a reconstruction loss $\mathcal{L}_{\text{Mask}} = \text{MSE}(M_{\text{pred}}, M_{\text{gt}})$ to optimize mask reconstruction, and a length regularization term $\mathcal{L}_{\text{Length}} = \lambda\hat{L}$ to encourage shorter token sequences, balancing

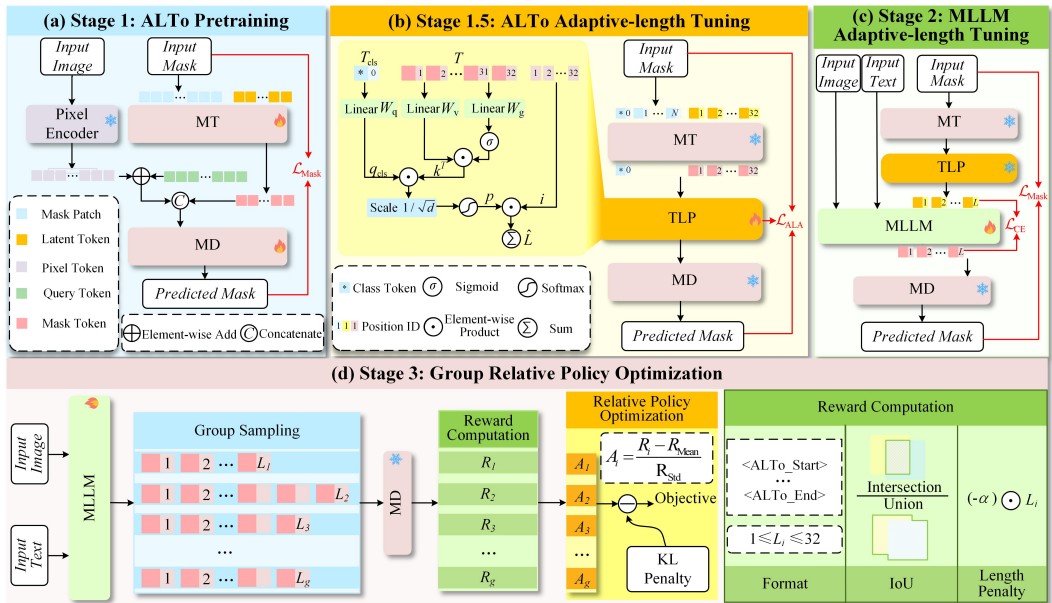

Figure 3: Training recipes for ALTo and ALToLLM. (a) **ALTo Pretraining**: Joint training of mask tokenizer (MT) and de-tokenizer (MD); (b) **Adaptive-length Prediction**: Training only the token length predictor (TLP); (c) **Multimodal Integration**: Exclusive training of MLLM with frozen ALTo for language-aware adaptation; (d) **Group Relative Policy Optimization**: Reinforcement learning for MLLM optimization. Input image in (b), (c) and (d) is processed identically to (a), omitted for visual clarity.

accuracy and efficiency while MT and MD are frozen. The final combined loss is $\mathcal{L}_{\mathrm{ALA}} = \mathcal{L}_{\mathrm{Mask}} + \mathcal{L}_{\mathrm{Length}}$. Further details about the length supervision design are provided in the Appendix. A.

### 3.3 ALToLLM

ALToLLM is built naturally on MLLM architecture, and learned by supervised fine-tuning (SFT) and group relative policy optimization (GRPO).

During SFT (stage 2), ALToLLM receives adaptive-length mask tokens provided by the frozen ALTo module, along with text and image tokens. ALToLLM is supervised using two objectives: a cross-entropy loss $\mathcal{L}_{\mathrm{ce}}$ for next-token prediction across the multimodal sequence, and a mask prediction accuracy loss $\mathcal{L}_{\mathrm{mask}}$, which combines binary cross-entropy loss and dice loss to ensure precise mask reconstruction. This dual-objective training enables ALToLLM to effectively align textual and visual information, and to autoregressively generate both language and adaptive-length mask tokens, which shows the effectiveness of ALTo.

We employ GRPO in stage 3 to adjust trade-off preferences flexibly based on the model after stage 2.

*1) Group sampling:* For each input consisting of an image and text, we sample $g$ multimodal responses from ALToLLM. A valid $i$-th sample must contain the following token sequence, where $L_i$ denotes the adaptive length:

$$\texttt{<ALTo\_Start>}\underbrace{\texttt{<TOK}_1\texttt{>}\cdots\texttt{<TOK}_{L_i}\texttt{>}}_{L_i \text{ tokens}}\texttt{<ALTo\_End>}, \quad L_i \in \{1, \dots, 32\} \tag{1}$$

*2) Reward computation:* The composite reward $R_i$ for the $i$-th sample consists of three components:

$$R_i = \underbrace{\mathbb{I}_{\mathrm{format}}}_{R_{\mathrm{valid}}} + \underbrace{\mathrm{IoU}}_{R_{\mathrm{accuracy}}} - \underbrace{\alpha L_i}_{R_{\mathrm{efficiency}}}, \tag{2}$$

where $R_{\mathrm{valid}}$ is $1$ if the sample strictly follows the format in Eq. 1, and $0$ otherwise. $R_{\mathrm{accuracy}}$ is the intersection-over-union (IoU) score between the predicted and ground truth masks, with the predicted mask reconstructed by the MD using the adaptive mask tokens; it vanishes to $0$ if any responses do

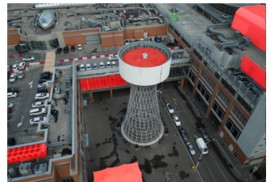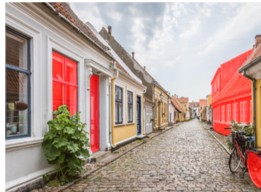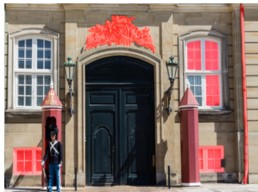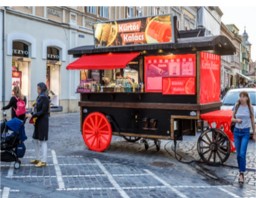

Figure 4: Examples from the Multi-Target-SA1B dataset.

Table 2: Performance comparison on gRefCOCO. We report cIoU, gIoU and average token length. FT indicates fine-tuning on referring expression data.

| Method | val | | | testA | | | testB | | |
|---|---|---|---|---|---|---|---|---|---|
| | cIoU | gIoU | Length | cIoU | gIoU | Length | cIoU | gIoU | Length |
| LISA-7B [27] | 38.7 | 32.2 | - | 52.6 | 48.5 | - | 44.8 | 39.7 | - |
| LISA-7B (FT) [27] | 61.8 | 61.6 | - | 68.5 | 66.3 | - | 60.6 | 58.8 | - |
| GSVA-7B [28] | 61.7 | 63.3 | - | 69.2 | 70.1 | - | 60.3 | 61.3 | - |
| GSVA-7B (FT) [28] | 63.3 | 66.5 | - | 69.9 | 71.1 | - | 60.5 | 62.2 | - |
| GroundHog-7B [51] | - | 66.7 | - | - | - | - | - | - | - |
| SAM4MLLM-8B [52] | 67.8 | 71.9 | - | 72.2 | 74.2 | - | 63.4 | 65.3 | - |
| UniRES++ [53] | 69.9 | 74.4 | - | 74.5 | 76.0 | - | 66.6 | 69.8 | - |
| LMM$_{\text{HiMTok}}$-8B [16] | 66.8 | 68.7 | 32 | 68.6 | 67.6 | 32 | 65.8 | 64.1 | 32 |
| LMM$_{\text{HiMTok}}$-8B (FT) [16] | 70.4 | 72.1 | 32 | 74.9 | 73.5 | 32 | 72.0 | 71.7 | 32 |
| ALToLLM-8B (FL) | 74.8 | 77.6 | 32 | 78.5 | 78.7 | 32 | 76.4 | 76.7 | 32 |
| ALToLLM-8B (AL) | **75.4** | **78.0** | 17.5 | **78.8** | **78.9** | 19.4 | **76.6** | **76.9** | 17.3 |

not conform to the correct format. $R_{\text{efficiency}}$ is a linear penalty $-\alpha L_i$ proportional to the token length $L_i \in \{1, \ldots, 32\}$, scaled by the trade-off parameter $\alpha \geq 0$.

*3) Relative policy optimization:* We optimize the policy using the clipped objective $\mathcal{J}_{\text{GRPO}}(\theta)$:

$$\mathbb{E}\left[\frac{1}{g}\sum_{i=1}^{g}\min\left(\frac{\pi_\theta}{\pi_{\text{old}}}A_i, \text{clip}\left(\frac{\pi_\theta}{\pi_{\text{old}}}, 1-\epsilon, 1+\epsilon\right)A_i\right) - \beta D_{\text{KL}}(\pi_\theta \parallel \pi_{\text{ref}})\right], \qquad (3)$$

where $\pi_\theta$ is the current policy being optimized (parameterized by $\theta$), $\pi_{\text{old}}$ is the policy before the update (used for importance sampling), $\pi_{\text{ref}}$ is the reference policy (typically the initial supervised policy), $A_i = (R_i - R_{\text{mean}})/R_{\text{std}}$ is the normalized advantage computed within each group, $D_{\text{KL}}$ is the Kullback-Leibler (KL) divergence enforcing policy stability, $\epsilon$ is the clip range (usually 0.1-0.3) controlling update aggressiveness, and $\beta$ is the KL penalty coefficient balancing exploration and constraint.

# 4 Experiments

## 4.1 Experimental settings

**Datasets**. For stages 1 and 1.5, we construct the training and validation sets of Multi-Target-SA1B from the SA1B dataset by randomly selecting multiple masks from all annotations for each image. Examples from Multi-Target-SA1B are shown in Fig. 4. This approach yields complex multi-target masks, facilitating the learning of expressive mask representations by ALTo. For stage 2, we used all HiMTok and Multi-Target-SA1B datasets for SFT. For Multi-Target-SA1B, we input the bounding boxes of all targets as "`<box>[[],[],...]</box>`". To ensure that the model supports both fixed-length and adaptive-length prompts, we randomly assign half of the data to each prompt type, as detailed in the Appendix. B. For stage 3, we use Multi-Target-SA1B, the RefCOCO series [48, 49], and gRefCOCO [50] to maintain complex mask representation and language understanding during RL.

**Implementation details**. ALTo processes input and reconstructs masks at $256 \times 256$ resolution. During training and inference, the MLLM processes images at $448 \times 448$, while the pixel encoder encodes image at $1024 \times 1024$. In stage 1, MT and MD are initialized from TiTok-L-32 [11] with codebook size of 1024, and the pixel encoder is initialized from SAM-ViT-L [54]. In stage 1.5, the feature dimension of TLP is set to 1024, consistent with MT. The length penalty coefficient is set to 0.0001, 0.001, 0.01, or 0.1, among which 0.01 is found to be optimal in subsequent experiments and is chosen for later stages. In stage 2, ALToLLM-8B is initialized from InternVL-2.5-8B [46]. Stage 3 trains the RL model based on the stage 2 checkpoint, with the length penalty set to 1e-2, 5e-3,

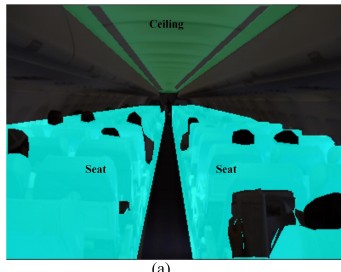 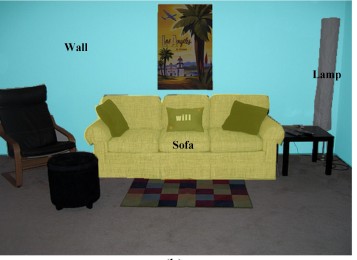 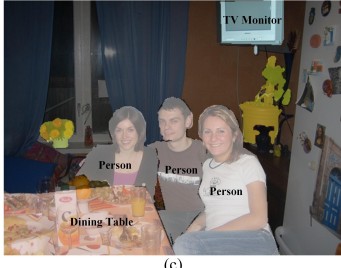

| (a) | (b) | (c) |

Figure 5: Examples from the constructed multi-class version of open-vocabulary segmentation datasets. (a) ADE20K (A-150); (b) PASCAL Context59 (PC-59); (c) PASCAL VOC 20 (PAS-20).

3e-3, 2e-3, 1e-3, or 1e-4, which are compared in later experiments. The KL penalty is set to 1e-3. We sample 12 group responses with a temperature of 1 and top-k of 10. Stages 1, 1.5, and 3 are trained on 8×A100 GPUs (80GB each), and stage 2 on 16×A100 GPUs. Training durations are: stage 1 for 2 days, stage 1.5 for 3 hours, stage 2 for 5 days, and stage 3 for 5 hours.

## 4.2 Comparative results

We evaluate ALToLLM-8B on the following tasks, considering two variants in SFT: (1) a fixed-length version (ALToLLM-8B (FL)) using 32 mask tokens, and (2) an adaptive-length version (ALToLLM-8B (AL)) that dynamically generates 1–32 mask tokens.

**Generalized referring expression segmentation with multiple targets**. We evaluate ALToLLM-8B on the generalized referring expression segmentation task (gRefCOCO [50]) and achieve state-of-the-art performance, as shown in Table 2. This task requires language-guided segmentation with multi-target referring expressions, demonstrating our model's ability to learn complex mask representations. Compared to the fixed-length variant (ALToLLM-8B (FL)), the adaptive-length variant (ALToLLM-8B (AL)) achieves higher performance and significantly reduces average token length, indicating improved segmentation accuracy and token efficiency. The superior performance of the adaptive-length approach may be attributed to its ability to use only the necessary tokens for simple masks, avoiding the noise introduced by redundant tokens. We also compare the average generation time per sample of the two variants. As shown in Table 3, the adaptive length variant achieves a consistently shorter generation time in all splits.

**Referring expression segmentation**. We further evaluate ALToLLM-8B on referring expression segmentation, a single-target version of the generalized task. Experiments are conducted on three standard benchmarks: RefCOCO [48], RefCOCO+ [48], and RefCOCOg [49]. As shown in Table 4, ALToLLM-8B achieves state-of-the-art results across all datasets.

**Multi-granularity segmentation**. We evaluate ALToLLM-8B on RefCOCOm [53], a multi-granularity referring segmentation dataset containing both part-level and object-level referring expressions. As shown in Table 5, ALToLLM-8B (AL) achieves the best performance.

Table 3: Comparison of average generation time per sample (in seconds) between fixed-length and adaptive-length variants on gRefCOCO. Generation time is measured on a single A100 GPU with batch size 1.

| Method | val | testA | testB |
|---|---|---|---|
| ALToLLM-8B (FL) | 1.079 | 1.079 | 1.076 |
| ALToLLM-8B (AL) | 0.710 | 0.753 | 0.669 |

**Multi-class open-vocabulary segmentation**. To demonstrate our model's ability to segment multiple and complex targets in open-vocabulary scenarios, we construct a multi-class version of open-vocabulary segmentation datasets by randomly merging annotations from several classes, including ADE20K (A-150) [62], PASCAL Context59 (PC-59) [63], and PASCAL VOC 20 (PAS-20) [64], as shown in Fig. 5. We reproduce the inference pipelines for LISA [27] and M²SA [56] for comparison. As shown in Table 6, ALToLLM-8B achieves state-of-the-art results.

## 4.3 Adaptive-length preference adjustment via reinforcement learning

By tuning the length penalty in the reward function, we can flexibly control the model's preference for adaptive token lengths while maintaining high IoU within a few hundred training steps by GRPO. As the average adaptive length decreases, the generation entropy also decreases, indicating that later

Table 4: Performance comparison on RefCOCO, RefCOCO+, and RefCOCOg. We report cIoU. FT indicates fine-tuning on referring expression data.

| Methods | RefCOCO | | | RefCOCO+ | | | RefCOCOg | |
|---|---|---|---|---|---|---|---|---|
| | val | testA | testB | val | testA | testB | val (U) | test (U) |
| Text4Seg-InternVL2-8B [25] | 79.2 | 81.7 | 75.6 | 72.8 | 77.9 | 66.5 | 74.0 | 75.3 |
| PolyFormer - B [55] | 74.8 | 76.6 | 71.1 | 67.6 | 72.9 | 59.3 | 67.8 | 69.1 |
| VistaLLM - 7B [33] | 74.5 | 76.0 | 72.7 | 69.1 | 73.7 | 64.0 | 69.0 | 70.9 |
| LISA - 7B [27] | 74.1 | 76.5 | 71.1 | 62.4 | 67.4 | 56.5 | 66.4 | 68.4 |
| LISA - 7B (FT) [27] | 74.9 | 79.1 | 72.3 | 65.1 | 70.8 | 58.1 | 67.9 | 70.6 |
| PixelLM - 7B [30] | 73.0 | 76.5 | 68.2 | 66.3 | 71.7 | 58.3 | 69.3 | 70.5 |
| GSVA - 7B [28] | 76.4 | 77.4 | 72.8 | 64.5 | 67.7 | 58.6 | 71.1 | 72.0 |
| GSVA - 7B (FT) [28] | 77.2 | 78.9 | 73.5 | 65.9 | 69.6 | 59.8 | 72.7 | 73.3 |
| PSALM [31] | 83.6 | 84.7 | 81.6 | 72.9 | 75.5 | 70.1 | 73.8 | 74.4 |
| GLaMM [29] | 79.5 | 83.2 | 76.9 | 72.6 | 78.7 | 64.6 | 74.2 | 74.9 |
| GroundHog - 7B [51] | 78.5 | 79.9 | 75.7 | 70.5 | 75.0 | 64.9 | 74.1 | 74.6 |
| SAM4MLLM - 8B [52] | 79.8 | 82.7 | 74.7 | 74.6 | 80.0 | 67.2 | 75.5 | 76.4 |
| M²SA - 7B [56] | 74.0 | 76.8 | 69.7 | 63.1 | 67.2 | 56.1 | 67.0 | 68.3 |
| AnyRef [57] | 74.1 | 75.5 | 70.8 | 64.1 | 68.7 | 57.5 | 68.1 | 69.9 |
| AnyRef (FT) [57] | 76.9 | 79.9 | 74.2 | 70.3 | 73.5 | 61.8 | 70.0 | 70.7 |
| SegAgent - LLaVA+SAM [58] | 79.2 | 81.4 | 75.7 | 71.5 | 76.7 | 65.4 | 74.8 | 74.9 |
| SegAgent - Qwen+SAM [58] | 78.0 | 80.3 | 75.0 | 70.9 | 75.5 | 65.8 | 74.5 | 74.6 |
| SegAgent - LLaVA+SClick [58] | 77.8 | 80.0 | 74.1 | 66.7 | 71.2 | 59.9 | 70.5 | 71.3 |
| SegAgent - Qwen+SClick [58] | 79.7 | 81.4 | 76.6 | 72.5 | 75.8 | 66.9 | 75.1 | 75.2 |
| Seg-Zero-7B [45] | - | 80.3 | - | - | 76.2 | - | - | 73.6 |
| LMM$_{HiMTok}$-8B [16] | 81.1 | 81.2 | 79.2 | 77.1 | 78.8 | 71.5 | 75.8 | 76.7 |
| LMM$_{HiMTok}$-8B (FT) [16] | 85.0 | 85.2 | 83.5 | 79.7 | 82.7 | 76.0 | 80.0 | 80.6 |
| ALToLLM-8B (FL) | 84.9 | 85.4 | 83.9 | **81.4** | **83.8** | **77.9** | 80.4 | 80.7 |
| ALToLLM-8B (AL) | **85.8** | **86.6** | **84.7** | 81.3 | **83.8** | 77.0 | **80.6** | **81.4** |

Table 5: Performance comparison on RefCOCOm. We report mIoU for part-level and object & part-level expressions. † indicates results reproduced by us using the official code and settings.

| Methods | val | | testA | | testB | |
|---|---|---|---|---|---|---|
| | Part | Obj & Part | Part | Obj & Part | Part | Obj & Part |
| X-Decoder [59] | 16.2 | 29.5 | 13.6 | 23.6 | 20.3 | 33.8 |
| SEEM [60] | 16.1 | 29.4 | 13.6 | 23.4 | 20.4 | 33.9 |
| UniRES [61] | 19.6 | 34.3 | 16.4 | 27.8 | 25.2 | 41.7 |
| LISA-7B [27] | 21.3 | 34.3 | 18.5 | 28.6 | 25.7 | 40.1 |
| GSVA-7B [28] | 11.4 | 23.1 | 9.2 | 19.2 | 16.8 | 28.2 |
| GLaMM [29] | 21.4 | 35.3 | 18.6 | 29.5 | 26.9 | 41.1 |
| M²SA-7B [56] | 22.4 | 35.5 | 19.9 | 30.1 | 27.1 | 41.4 |
| LMM$_{HiMTok}$-8B† [16] | 23.4 | 37.3 | 20.7 | 31.5 | 28.3 | 45.0 |
| ALToLLM-8B (FL) | **25.5** | **39.2** | **22.6** | **33.3** | **30.3** | **46.7** |
| ALToLLM-8B (AL) | **25.5** | 39.1 | **22.6** | 33.2 | 30.2 | 46.5 |

tokens are associated with higher uncertainty. This trend is visualized in Fig. 6. To quantify the token savings achieved by adaptive-length tokens at various IoU levels, we compare six models trained with different length penalties in stage 3, alongside a fixed-length baseline from stage 2. Validation is conducted on Multi-Target-SA1B for complex mask representation and gRefCOCO [50] for language understanding. For zero-shot evaluation, we use the multi-class A-150 dataset, which is not seen during stage 3 training. As shown in Fig. 7, adaptive-length models consistently save more than 10 tokens at the same IoU level, and this advantage persists even in zero-shot scenarios.

## 4.4 Ablation Studies

**Ablation on ALTo.** We first verify the necessity of the pixel encoder for reconstructing complex masks. As shown in Table 7, removing the pixel encoder leads to a substantial drop in reconstruction gIoU on the Multi-Target-SA1B validation set, confirming that pixel-level visual features are essential for capturing fine-grained mask details. We also study the effect of different penalty coefficients during Stage 1.5 training on adaptive-length mask prediction, as illustrated in Fig. 8. The results show that a coefficient of 0.01 strikes an optimal balance: it maintains high mask quality (in terms of IoU) while enabling a diverse range of predicted token lengths, as evidenced by high standard deviation and entropy in output lengths. We therefore adopt this value for Stage 2 training.

**Ablation on ALToLLM.** To better understand the key components driving ALToLLM's performance gains, we conduct comprehensive ablation studies on gRefCOCO [50], with results summarized in Table 8. Our analysis reveals that ALToLLM's improvements in mask quality primarily stem

Table 6: Performance comparison on multi-class open-vocabulary segmentation. We report gIoU.

| Method | A-150 | PC-59 | PAS-20 |
|---|---|---|---|
| LISA-7B [27] | 39.1 | 53.3 | 67.8 |
| M²SA-7B [56] | 63.9 | 74.1 | 79.5 |
| ALToLLM-8B (FL) | 65.6 | 75.2 | **81.2** |
| ALToLLM-8B (AL) | **65.8** | **75.4** | 81.1 |

Table 7: Ablation study of the pixel encoder on the Multi-Target-SA1B validation dataset. We report gIoU.

| Method | Pixel Encoder | gIoU |
|---|---|---|
| ALTo (Ours) | ✓ | 94.4 |
| ALTo (Ours) | ✗ | 91.9 |

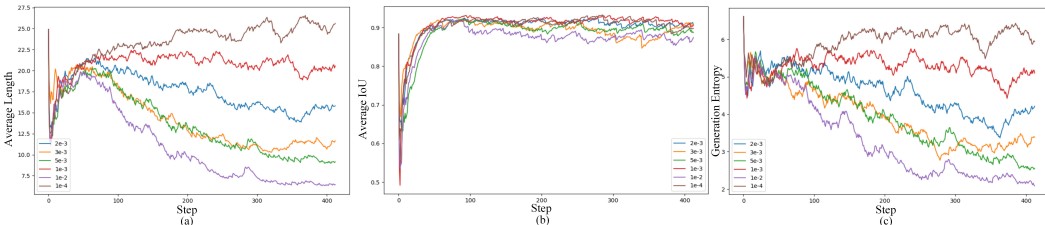

Figure 6: Metrics of sampled responses during GRPO training. (a) Average token length, (b) Average IoU, (c) Generation entropy.

from two factors: (1) pretraining on Multi-Target-SA1B, a dataset containing scenes with multiple and structurally complex objects, and (2) the pixel encoder, which provides high-resolution visual cues that refine mask boundary reconstruction. Moreover, the adaptive-length setting in SFT not only improves token efficiency—reducing the average output length by approximately 50%—but also slightly enhances mask quality. This demonstrates that adaptive token generation effectively eliminates redundancy while preserving, and even improving, mask quality. Finally, GRPO contributes marginally to absolute performance metrics but helps the model learn an effective trade-off between mask quality and token efficiency.

## 5 Conclusions

We present ALToLLM, an innovative framework that dynamically adapts the number of mask tokens according to object complexity. ALToLLM approaches mask tokens as a visual language system, where our ALTo intelligently determines the optimal token count for each object. Furthermore, by integrating ALTo with MLLMs, the system can interpret linguistic descriptions and correspondingly adjust token allocation. Extensive experiments demonstrate state-of-the-art performance on RefCOCO [48], RefCOCO+ [48], RefCOCOg [49], RefCOCOm [53], and gRefCOCO [50] benchmarks, validating our approach's effectiveness in aligning linguistic expressions with adaptive mask tokenization.

While ALTo effectively handles most segmentation tasks with adaptive token lengths (1-32 tokens), two key directions merit further exploration. First, our approach requires multiple training stages, which increases the engineering complexity. A simpler training pipeline is desirable for future appli-

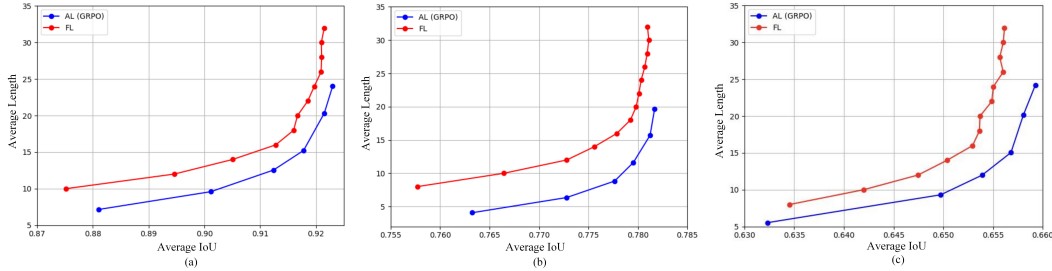

Figure 7: Comparison of token cost between fixed-length and adaptive-length models with different length preferences. FL denotes the fixed-length model from stage 2. AL denotes stage 3 models trained with different length penalties (from left to right: 1e-2, 5e-3, 3e-3, 2e-3, 1e-3, 1e-4). (a) Multi-Target-SA1B val, (b) gRefCOCO val, (c) Multi-Class A-150 (zero-shot).

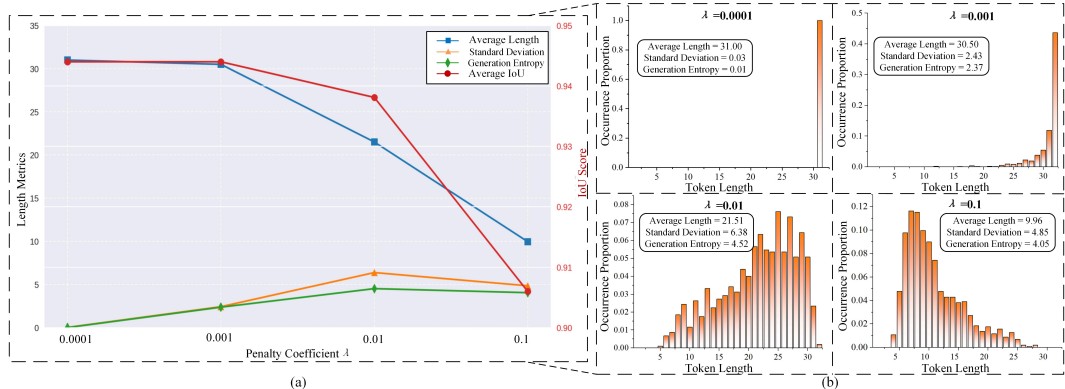

Figure 8: Analysis of the penalty coefficient in stage 1.5. (a) Length metrics for different penalty coefficients. (b) Distribution of length metrics for penalty coefficients 0.0001, 0.001, 0.01, and 0.1.

Table 8: Ablation study for ALToLLM on gRefCOCO validation dataset.

| Components | | | | Metrics | | |
| --- | --- | --- | --- | --- | --- | --- |
| Multi-Target-SA1B | Pixel Encoder | AL(SFT) | GRPO | cIoU | gIoU | Avg. Length |
| | | | | 70.4 | 72.1 | 32.0 |
| ✓ | | | | 72.8 | 75.6 | 32.0 |
| ✓ | ✓ | | | 74.8 | 77.6 | 32.0 |
| ✓ | ✓ | ✓ | | 75.4 | 78.0 | 17.5 |
| ✓ | ✓ | ✓ | ✓ | **75.5** | **78.1** | **15.7** |

cations. Second, though our pipeline is designed to be modality-agnostic (treating mask tokenization as a special case of image tokenization), our experiments currently focus on mask validation. Future work will extend ALTo to RGB image tokenization and validate its effectiveness across more general vision tasks.

# Acknowledgment

This research was supported by the Key Scientific Research Program of Hangzhou Municipal Bureau of Science and Technology grants 2025SZD1A01.

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

# A  Details of TLP

The Adaptive-Length Tokenizer (ALTo) utilizes a token length predictor (TLP) to dynamically determine the number of tokens required for each mask. In this section, we provide further mathematical details and training insights to clarify the **differentiable token chunking** strategy.

The overall loss function consists of a reconstruction loss and a length regularization term:

$$\mathcal{L}_{\text{ALA}} = \mathcal{L}_{\text{Mask}} + \mathcal{L}_{\text{Length}},$$

where $\mathcal{L}_{\text{Length}} = \lambda \hat{L}$ penalizes longer token sequences. The gradient of the loss is given by

$$\nabla \mathcal{L}_{\text{ALA}} = \nabla \mathcal{L}_{\text{mask}} + \nabla \mathcal{L}_{\text{length}}$$
$$= \sum_i \frac{\partial \mathcal{L}_{\text{mask}}}{\partial \hat{T}_i} \nabla \hat{T}_i + \lambda \sum_i i \nabla p_i$$

If we directly chunk the token sequence, such as $\hat{T} = H \odot T$, we obtain $\nabla \hat{T} = 0$ because $H$ is non-differentiable and $T$ is generated by a frozen mask tokenizer. As a result, the gradient of $\mathcal{L}_{\text{mask}}$ cannot be backpropagated to the TLP, and only the token length is optimized to be minimal.

To address this limitation, we introduce a differentiable token chunking strategy. Specifically, we use the cumulative probability $P_i = 1 - \sum_{j < i} p_j$ to represent the probability that the $i$-th token is used, and $\hat{P}$ denotes the detached version of $P$. We then construct a soft token chunking as $\hat{T} = (P - \hat{P} + H) \odot T$. In this way, we have $\nabla \hat{T} = \nabla P \odot T$. The overall gradient is given by

$$\nabla \mathcal{L}_{\text{ALA}} = \nabla \mathcal{L}_{\text{mask}} + \nabla \mathcal{L}_{\text{length}}$$
$$= \sum_i \frac{\partial \mathcal{L}_{\text{mask}}}{\partial \hat{T}_i} P_i T_i + \lambda \sum_i i \nabla p_i$$
$$= \sum_i \frac{\partial \mathcal{L}_{\text{mask}}}{\partial \hat{T}_i} T_i \sum_{j \geq i} \nabla p_j + \lambda \sum_i i \nabla p_i$$
$$= \sum_i \nabla p_i \sum_{j \leq i} \frac{\partial \mathcal{L}_{\text{mask}}}{\partial \hat{T}_j} T_j + \lambda \sum_i i \nabla p_i$$
$$= \sum_i \nabla p_i \left( \sum_{j \leq i} \frac{\partial \mathcal{L}_{\text{mask}}}{\partial \hat{T}_j} T_j + \lambda i \right)$$

for the best predict length $\hat{L} = k$, there is $(\sum_{j \leq k} \frac{\partial \mathcal{L}_{\text{mask}}}{\partial \hat{T}_j} T_j + \lambda k) < (\sum_{j \leq k-1} \frac{\partial \mathcal{L}_{\text{mask}}}{\partial \hat{T}_j} T_j + \lambda(k-1))$ and $(\sum_{j \leq k} \frac{\partial \mathcal{L}_{\text{mask}}}{\partial \hat{T}_j} T_j + \lambda k) < (\sum_{j \leq k+1} \frac{\partial \mathcal{L}_{\text{mask}}}{\partial \hat{T}_j} T_j + \lambda(k+1))$ , which means

$$-\frac{\partial \mathcal{L}_{\text{mask}}}{\partial \hat{T}_{k+1}} T_{k+1} < \lambda < -\frac{\partial \mathcal{L}_{\text{mask}}}{\partial \hat{T}_k} T_k$$

This form shows that the model is encouraged to select the minimal number of tokens that still achieve high reconstruction quality, as the regularization term $\lambda$ acts as a threshold for including additional tokens.

Intuitively, the model will only increase the predicted token length if the marginal gain in reconstruction quality outweighs the regularization penalty. This mechanism is analogous to a reward-cost trade-off in reinforcement learning, where the "reward" for including an additional token must exceed the cost $\lambda$.

# B  Prompt design

We prepared different prompt templates for instruction tuning on adaptive-length and fixed-length segmentation. Tabs 9 and 10 are the templates for adaptive-length ALToLLM tuning, and Tabs 11 and 12 are the templates for fixed-length version.

Table 9: Templates of instruction for adaptive-length segmentation.

- "Segment <ref>{}</ref> by adaptive length."

- "Create a mask for <ref>{}</ref> by adaptive length."

- "Generate a mask for <ref>{}</ref> by adaptive length."

- "Do segmentation for <ref>{}</ref> by adaptive length."

- "Please give the mask for <ref>{}</ref> by adaptive length."

- "What is the mask for <ref>{}</ref> by adaptive length?"

- "Can you segment <ref>{}</ref> by adaptive length?"

Table 10: Templates of response for adaptive-length segmentation.

- "The adaptive mask appears at {}."

- "The adaptive mask is created as {}."

- "I can generate the adaptive mask at {}."

- "The adaptive mask is {}."

- "I can give the adaptive mask at {}."

- "Its adaptive mask located at {}."

- "Sure, the adaptive mask is {}."

Table 11: Templates of instruction for fixed-length segmentation.

- "Segment <ref>{}</ref>"

- "Create a mask for <ref>{}</ref>"

- "Generate a mask for <ref>{}</ref>"

- "Do segmentation for <ref>{}</ref>"

- "Please give the mask for <ref>{}</ref>"

- "What is the mask for <ref>{}</ref>?"

- "Can you segment <ref>{}</ref>?"

Table 12: Templates of response for fixed-length segmentation.

- "The mask appears at {}."

- "The mask is created as {}."

- "I can generate the mask at {}."

- "The mask is {}."

- "I can give the mask at {}."

- "Its mask located at {}."

- "Sure, the mask is {}."

# C    Results on reasoning segmentation

We evaluate ALToLLM-8B on the ReasonSeg benchmark [27], which demands complex visual reasoning for accurate segmentation. As shown in Table 13, our method achieves consistently superior performance in both cIoU and gIoU, demonstrating enhanced reasoning capabilities over the baseline.

Table 13: Reasoning segmentation results on ReasonSeg validation dataset.

| Methods | cIoU | gIoU |
|---|---|---|
| LISA-7B [27] | 46.0 | 44.4 |
| LISA-7B (ft) [27] | 54.0 | 52.9 |
| SAM4MLLM-8B [52] | 60.4 | 58.4 |
| HiMTok [16] | 67.0 | 60.7 |
| ALToLLM-8B | **67.3** | **62.8** |

# D    Results on region understanding

Following the setup of GLaMM [29], we assess Region-Level Captioning on the RefCOCOg dataset [49]. Our method outperforms GLaMM in both METEOR and CIDEr metrics, demonstrating enhanced ability to generate semantically accurate and descriptive captions for localized image regions.

Table 14: Region-level captioning results on RefCOCOg.

| Methods | METEOR | CIDEr |
|---|---|---|
| GLaMM | 16.2 | 106.0 |
| ALToLLM-8B | **16.5** | **110.1** |

# E    Application examples

Fig. 9 and Fig. 10 illustrates examples of ALToLLM in practical applications. As demonstrated, ALToLLM effectively segments the target object referred to by the user by leveraging adaptive-length mask tokens.

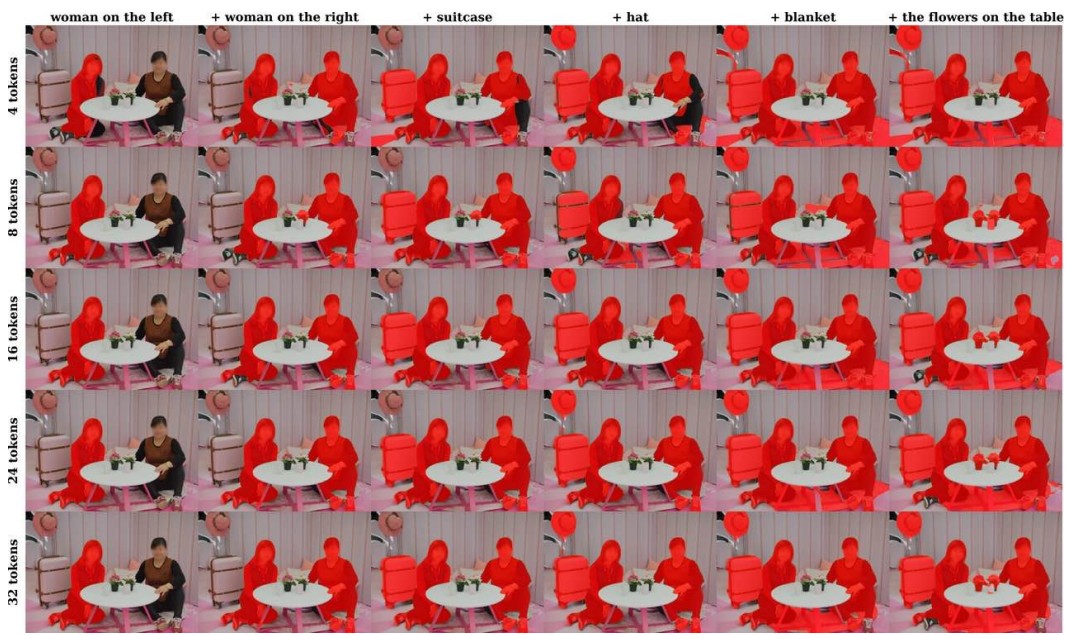

Figure 9: An example of complex scenarios

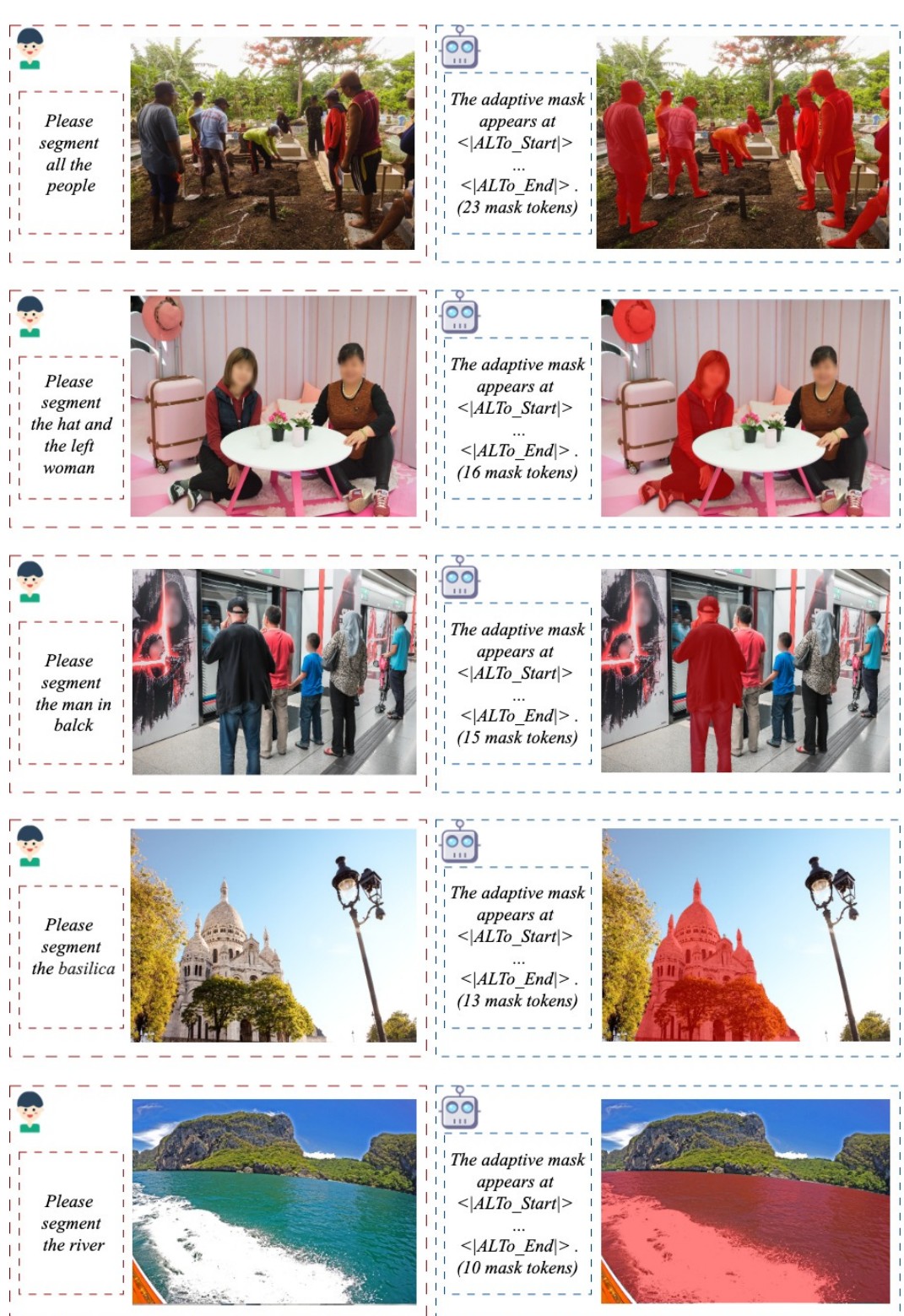

Figure 10: Examples of ALToLLM with adaptive-length segmentation.

