# OpenReview forum: "ALTo: Adaptive-Length Tokenizer for Autoregressive Mask Generation"
_NeurIPS.cc/2025/Conference — NeurIPS 2025 poster_

### Official Review · Reviewer_7PTq · 2025-06-07

**Clarity:** 3
**Significance:** 3
**Originality:** 3
**Rating:** 4
**Confidence:** 3

**Summary:**

This paper introduces ALTo (Adaptive-Length Tokenizer) and ALToLLM, a framework that enables MLLMs to generate adaptive-length mask tokens for object segmentation. ALTo uses a Token Length Predictor (TLP) to determine the optimal number of tokens based on object complexity, combined with a length regularization term and differentiable token chunking. ALToLLM integrates this into an MLLM, using Group Relative Policy Optimization (GRPO) to balance mask quality and efficiency. Experiments show ALToLLM achieves state-of-the-art results on segmentation benchmarks while reducing token usage and generation time.

**Questions:**

Please see weaknesses.

**Ethical Concerns:**

["NO or VERY MINOR ethics concerns only"]

**Final Justification:**

The authors' rebuttal has addressed my concerns and I tend to accept this paper.

**Limitations:**

The authors state in the checklist that "The limitations of our work are discussed in the Supplementary Material." However, I could not find a corresponding section discussing the limitations in the appendix or supplementary material.

**Paper Formatting Concerns:**

No major formatting issuses.

**Quality:**

3

**Strengths And Weaknesses:**

Strengths:

1. ALTo allows the model to automatically decide the token length for mask generation, which sounds reasonable because the number of tokens required to represent fine-grained edge shapes can vary drastically depending on their complexity.
2. This work utilizes GRPO to dynamic control over token length, which balances mask quality and computational efficiency.
3. ALToLLM achieves state-of-the-art performance with adaptive token usage across various object segmentation benchmarks.

Weaknesses:
1. The paper does not provide a quantitative evaluation regarding the effectiveness of the proposed adaptive length tokenizer (ALTo); for example, how does ALTo handle extremely complex objects that might require more than 32 tokens?
2. The paper claims that "We employ GRPO to further adjust specific preferences on the trade-off between mask quality and token efficiency." However, it does not seem to provide a quantitative analysis of the impact of GRPO on mask quality and token efficiency.
3. In Figure 1, the first three cases presented involve relatively simple segmentation scenarios, where increasing the number of mask tokens does not seem to have a significant effect. The authors should provide more complex scenarios to better demonstrate the effectiveness of the adaptive token approach.

---

> ### Author Rebuttal · Authors · 2025-07-30
>
> We thank the reviewer for insightful and constructive feedback on our work.
>
> >  **W1.1: Provide a quantitative evaluation**
>
> **A:** We clarify that the effectiveness of the proposed adaptive-length tokenizer (ALTo) is quantitatively evaluated on our MultiSA1B dataset, which contains complex shapes and multi-object masks, as shown in **Figure 8**. The evaluation includes multiple settings of the length penalty coefficient. With a coefficient of 0.01—which is the value selected for our final model—we achieve an average output length of 21.51 (standard deviation: 6.38) and a high average mask IoU of 0.938, demonstrating the strong effectiveness of our approach in balancing token efficiency and segmentation accuracy.
>
> Beyond the tokenizer itself, the advantages of our method are further supported by the state-of-the-art performance and significantly reduced token usage of ALToLLM-8B in **Table 2**.
>
> >  **W1.2: How does ALTo handle extremely complex objects that might require more than 32 tokens**
>
> **A:**  **Theoretical Basis**. The maximum token length of 32 in ALTo is a deliberate design choice, inspired by the finding in TiTok [11] that "an image is worth 32 tokens," which demonstrates that a compact 1D sequence of 32 tokens can effectively represent complex visual information.  **Empirical Validation**. (1) Our extensive experiments, including those on the highly complex Multi-Target-SA1B dataset, have shown that 32 tokens are sufficient to achieve high-quality segmentation.  (2) As shown in Figure 8, our selected penalty coefficient (0.01) naturally leads to token lengths below 32 for most samples, with fewer than 0.5% of cases reaching the maximum. (3) As evidence in the response to W3, even for complex multi-object scenes, such as an image containing six distinct objects, our model could still mask all objects within 32 tokens. This design ensures a favorable balance between representational capacity and computational efficiency.
>
> >  **W2: Provide a quantitative analysis of the impact of GRPO**
>
> **A:** In fact, **Figure 7** illustrates how GRPO influences mask quality and token efficiency. The x-axis shows IoU, while the y-axis represents the average sequence length. Blue points correspond to ALToLLM variants trained with different GRPO penalty coefficients, whereas red points indicate the fixed-length baseline. When comparing models with similar IoU (i.e., at the same x-coordinate), ALToLLM achieves comparable segmentation performance while reducing token usage by more than 5 tokens per sample on average—demonstrating superior **mask quality and token efficiency**.
>
> Furthermore, as the length penalty coefficient in GRPO training decreases, mask quality improves at the expense of longer output sequences, confirming a controllable trade-off between accuracy and efficiency. This enables users to flexibly tailor the model’s behavior based on whether **mask quality or token efficiency** is prioritized in a specific application. As shown in **Figure 6**, this adaptation is achieved within just a few hundred training steps, supporting our claim in **L170** that GRPO provides **flexible** and efficient control over the model’s output characteristics.
>
> >  **W3: Provide more complex scenarios**
>
> **A:** We sincerely appreciate the reviewer's insightful feedback regarding the demonstration of ALTo's adaptive capabilities. While our initial **Figure 1** illustrated the progressive token allocation for objects with increasing edge complexity (from 4 tokens for a simple moon to 17 tokens for complex land boundaries), we fully agree that more sophisticated multi-object scenarios would better showcase our method's advantages. To comprehensively address this point, we analyze a complex scene with sequentially added objects. The results demonstrate ALTo's dynamic scaling of token allocation: beginning with 7 tokens for a single woman, the count appropriately increases to 30 tokens when incorporating fine details like flowers on a table. This systematic evaluation confirms our method's ability to automatically adjust token usage based on both object quantity and geometric complexity.
> Due to the NeurIPS rebuttal constraints, we present these findings in tabular form below and will include the complete visual analysis in the supplementary material for the final version.
> | Object Description       | Number of Mask Tokens | Number of Objects |
> |--------------------------|-----------------------|-------------------|
> | the woman on the left    | 7                     | 1                 |
> | + the woman on the right  | 18                    | 2                 |
> | + suitcase                | 20                    | 3                 |
> | + hat                     | 22                    | 4                 |
> | + blanket                 | 25                    | 5                 |
> | + flower on the table     | 30                    | 6                 |
>
> >  **Limitations**
>
> **A:** We sincerely apologize for this oversight in the supplementary material. The "Limitations and Future Work" section was inadvertently omitted during the final formatting stage. We will immediately add this section to the supplementary material.
>
>  **Limitations and Future Work**
>
> While ALTo effectively handles most segmentation tasks with adaptive token lengths (1-32 tokens), two key directions merit further exploration.  First, our approach requires multiple training stages, which increases the engineering complexity. A simpler training pipeline is desirable for future applications. Second, though our pipeline is designed to be modality-agnostic (treating mask tokenization as a special case of image tokenization), our experiments currently focus on mask validation. Future work will extend ALTo to RGB image tokenization and validate its effectiveness across more general vision tasks.

---

> > ### Comment · Reviewer_7PTq · 2025-08-03
> >
> > The authors' response have addressed my concerns.

---

> > > ### Author Response · Authors · 2025-08-09
> > >
> > > Thank you! We're glad we could address your concerns.

---

### Official Review · Reviewer_bQXT · 2025-06-27

**Clarity:** 3
**Significance:** 2
**Originality:** 2
**Rating:** 4
**Confidence:** 4

**Summary:**

This paper proposes an adaptive method, ALTo,  for generating adaptive length tokens for the task of image segmentation. The length of the token sequences depends on the difficulty of the segmented objects. In addition, ALTo can be plugged into the Multimodal Large Language Model (MLLM) through SFT and RL, enabling the task of referring segmentation. Through the comprehensive experiment, the paper demonstrates the efficiency of the adaptive length tokens compared with the baselines: Achieving high segmentation results while being fast compared with the fixed-length token mechanism.

**Questions:**

- Restricted only to segmentation: Can this approach be extended to other tasks like Image understanding and region understanding, similar to GLaMM[1]? What is the performance of the adaptive tokenization for the Reasoning Segmentation task? Please refer to Sections 4.4 and 4.7 for metrics which is not Segmentation.
- Datasets used in the experiments: What is the difference in terms of problem setting between different types of datasets used in this paper? For example, what is the difference between gRefCOCO, RefCOCO, and RefCOCO+? The paper should include the difference between these types of datasets for the reader to understand.
- Ablation study: This paper should include the ablation study about the importance of different stages to the final results. For example, how important of RL phase compared with the results of the SFT phase only?
- In L153, what is $\hat{P}$? It is never defined.
- In L156, to encourage short token regularization, is $\mathcal{L}_{length}$ should be $\lambda L$ instead of $-\lambda L$.
- Why is the method used to compare with ALToLLM in Table 2 and Table 3 not consistent? The method list is not similar for both tables, making it harder to understand what this paper is trying to compare.
- What is the main factor that makes the difference between Fix-length ALTo with Fix-length tokenization compared with HimTok in Table 2 and 4?

Reference:
[1] Rasheed, Hanoona, et al. "Glamm: Pixel grounding large multimodal model." Proceedings of the IEEE/CVF Conference on Computer Vision and Pattern Recognition. 2024.

**Ethical Concerns:**

["NO or VERY MINOR ethics concerns only"]

**Final Justification:**

Based on the rebuttal, the authors have addressed my concerns. Based on the range of the application of the adaptive-length token representation, I will increase the score to 4.

**Limitations:**

See the question section.

**Paper Formatting Concerns:**

No concern about paper formatting

**Quality:**

2

**Strengths And Weaknesses:**

**Strength**: This paper tries to improve HimTok [1] in terms of the number of tokens for the problem of pixel understanding (segmentation) in the MMLLM model. The approach is to use an adaptive token strategy, which uses a different number of tokens for each segmentation task, depending on the difficulty of the segmentation. This approach is intuitive, and the experiment indicates the efficacy of their approach.

**Weakness**: Most of the parts are borrowed from the HimTok work, and only token adaptation is the main contribution of this work. However, the computation of the token adaptation at some point (straight-through estimator in L153) is not clear. The changes in terms of the number of tokens should not affect the results of the baselines MMLLM in some evaluation metrics besides segmentation, e.g., segmentation reasoning and general image understanding, which is not reported in this paper. Further ablation study is required to indicate the efficacy of each component (training phase) with the overall performance. However, the proposed approach is interesting for reducing the number of tokens used in HimTok.

[1] Wang, Tao, et al. "Himtok: Learning hierarchical mask tokens for image segmentation with large multimodal model." arXiv preprint arXiv:2503.13026 (2025).

---

> ### Author Rebuttal · Authors · 2025-07-30
>
> Thanks for the constructive comments on the intuitive, interesting and effective method.
>
>
> >  **W1: Most of the parts are borrowed from the HimTok work, and only token adaptation is the main contribution of this work**
>
> **A:** We would like to clarify that the proposed adaptive-length mask tokenizer and its integration into LMM -- resulting in ALToLLM -- are novel and non-trivial contributions, even though they are experimentally built upon the HiMTok architecture. The method introduces crucial innovations:
> - **The First Autonomous Adaptive-Length Mask Tokenizer**. As shown in **Table 1**, ALTo represents the first framework enabling models to dynamically determine visual mask token lengths (2-32) based on shape complexities.
> - **Adaptive-Length Token Generation in MLLM**.  ALToLLM enables adaptive mask token generation for (referring) image segmentation tasks through SFT and GRPO, with elaborately-designed learning objective and RL reward. We can balance the  quality and efficiency conviniently.
>
> This has the potential to be applied to a broader range of visual scenarios in the future.
>
> >  **Q1: Extended to other tasks**
>
> **A:** We appreciate the reviewer's suggestion to evaluate broader capabilities. Our experiments demonstrate ALToLLM's versatility across multiple vision-language tasks.
>
> (1) **Reasoning Segmentation**
> We evaluate our model on the *ReasonSeg* benchmark, which requires complex reasoning for segmentation. Our method achieves superior performance compared to HiMTok, indicating stronger reasoning capabilities. These improvements are consistent with findings on other segmentation benchmarks such as gRefCOCO, further validating the robustness of our approach.
>
> | Methods | cIoU | gIoU  |
> |---------|------|-------|
> | HiMTok  | 67.0 | 60.7  |
> | Ours    | **67.3** | **62.8**  |
>
> (2) **Region Understanding**
> Following the setup of GLaMM, we assess Region-Level Captioning on the *RefCOCOg* dataset. Our method outperforms GLaMM in both METEOR and CIDEr metrics, demonstrating enhanced ability to generate semantically accurate and descriptive captions for localized image regions.
> | Methods| METEOR | CIDEr  |
> |--------|--------|--------|
> | GLaMM  | 16.2   | 106.0  |
> | Ours   | **16.5**   | **110.1**  |
>
> (3) **Image Understanding**
> We further evaluate image captioning performance on *NoCaps* and *Flickr30K*, following the protocol of GLaMM. Two settings are considered: (i) the original setting, and (ii) a fine-tuned setting (denoted with †) using a small number of COCO training samples provided by GLaMM.
> - In the original setting, our model tends to generate more detailed and descriptive captions—often exceeding the length and specificity of ground-truth annotations—leading to higher SPICE scores but lower CIDEr values due to style mismatch. This reflects our model’s strong scene understanding and descriptive richness.
> - In the fine-tuned setting (†), our model successfully adapts to the concise captioning style typical of these benchmarks. On NoCaps, we achieve a CIDEr score of **113.6**, surpassing GLaMM, while maintaining competitive SPICE performance. This adaptability underscores the flexibility and broad applicability of ALToLLM across different captioning styles and domains.
>
> | Methods | NoCap          |            | Flickr30k     |            |
> |---------|----------------|------------|---------------|------------|
> |         | CIDEr          | SPICE      | CIDEr         | SPICE      |
> | GLaMM   | 106.8          | 15.8       | 95.3          | 18.8       |
> | Ours    | 61.7           | **17.1**       | 57.0          | **20.6**       |
> | Ours†   | **113.6**          | 14.8       | 82.8          | 18.0       |
>
>
>
> >  **Q2: The difference in terms of problem setting between different types of datasets**
>
> **A:** Thank you for raising this important point. We clarify the key differences in the problem settings across the benchmark datasets:
>
> - **RefCOCO [48]** focuses on basic referring expressions that rely on spatial cues (e.g., *"left woman"*).
> - **RefCOCO+ [48]** prohibits the use of absolute spatial terms, requiring models to understand attribute-based descriptions (e.g., *"woman in red"*).
> - **RefCOCOg [49]** emphasizes unambiguous expressions to distinguish between visually similar objects (e.g., *"woman who is touching her head"*).
> - **gRefCOCO [50]** includes generalized referring expressions, covering single-object, **multi-object**, and empty references (e.g., *"all women"*).
> - **RefCOCOm [53]** introduces **part-level segmentation**, requiring fine-grained understanding of object parts (e.g., *"woman's hat"*).
>
> This progression from simple to complex referring expressions systematically evaluates a model's ability to handle various linguistic and visual challenges. Among them, **gRefCOCO**, with its focus on multi-object references, is particularly relevant for assessing the effectiveness of our adaptive tokenization mechanism in complex, real-world scenes.
>
> > **Q3 & 7: Ablation studies contributing to the final results**
>
> **A:** Thank you for your interest in our ablation studies. Building upon the results on the *gRefCOCO val* set (**Table 2**), we further incorporate the GRPO results with a penalty coefficient of $1 \times 10^{-3}$ (as shown in **Figure 7**). We perform more ablation experiments on the following components:
> - **Multi-Target-SA1B Pretraining**: Utilizing our curated dataset containing complex and multi-object masks during Stage 1 pretraining.
> - **Pixel Encoder**: Incorporating a pixel-level encoder to provide fine-grained visual features to the mask de-tokenizer.
>
> The results are summarized below:
>
> | Methods           |          |            |     | Metrics           | | |
> |:-----------------:|:-------------:|:--------:|:-------------:|:-----:|:-----:|:-----------:|
> | Multi-Target-SA1B | Pixel Encoder | Adaptive Length | GRPO Training | cIoU  | gIoU  | Avg. Length |
> |                   |               |              |         | 70.4  | 72.1  |     32.0    |
> |         ✓         |               |               |        | 72.8  | 75.6  |     32.0    |
> |         ✓         |       ✓       |                |       | 74.8  | 77.6  |     32.0    |
> |         ✓         |       ✓       |           ✓    |       | 75.4  | 78.0  |     17.5    |
> |         ✓         |       ✓       |           ✓    |✓              | **75.5**  | **78.1**  |     **15.7**    |
>
> > *Q3: RL phase compared with the results of the SFT phase*
>
> *A:* As shown in Figure 7, the performance gain from the GRPO phase compared to the SFT phase (**Table 2**) is relatively modest in terms of absolute metrics (e.g., +0.1 cIoU). However, as noted in L170, the main advantage of GRPO lies in its **flexibility**—it allows users to control the trade-off between mask quality and token efficiency.
> By adjusting the length penalty coefficient during GRPO training, we observe that **reducing the penalty leads to higher mask quality at the cost of increased token usage**, while increasing the penalty yields more compact outputs with slightly lower mask fidelity. This level of controllability enables the model to be adapted to different application scenarios, depending on whether users prioritize high-quality masks or efficient token generation.
>
> > *Q7: Main factors contributing to the performance gains over HimTok in Tables 2 and 4*
>
> *A:* The table above shows that the improvements are primarily attributed to two key components:
> - (1) **Multi-Target-SA1B Pretraining**, which significantly enhances the model’s ability to generalize to complex scenes with multiple objects.
> - (2) **Pixel Encoder**, which provides fine-grained visual representations that enable more accurate reconstruction of detailed mask boundaries.
>
> Importantly, the performance improvement on *gRefCOCO* (**Table 2**) is notably larger than that observed on *RefCOCO*, *RefCOCO+*, and *RefCOCOg* (**Table 4**). This indicates that our model excels particularly well in **multi-object settings** (see **Q2**), which aligns with the design goals of the Multi-Target-SA1B pretraining and the fine-grained visual modeling enabled by the pixel encoder.
>
> >  **Q4 & 5: Problems on writing (L153&156)**
>
> **A:** Thank you for your careful comments on the writing issues. We will thoroughly review the manuscript.
> - In L153, $\hat{P}=P.detach()$, which is detached from the computation graph, so that $\hat{T}$ is differentiable with respect to $P$.
> - In L156, thanks for pointing out this typo — it is indeed $\mathcal{L}_{\text{Length}} = \lambda \hat{L}$.
>
> >  **Q6: The method list is not similar for Table 2 and Table 3**
>
> **A:** Different dimensions are compared using different method list in Tables 2 and 3, respectively.
> - It is natural to compare performance metrics (e.g., cIoU and gIoU) with many related works (method list in Tab. 2), but the average token length is only meaningful for methods following HiMTok paradigm (i.e., discrete mask token generation). Thus we only compare ALToLLM-8B (AL) with ALToLLM-8B (FL) and HiMTok on average token length in **Table 2**.
> - To better illustrate the efficiency gains brought by the adaptive token mechanism, we also report the corresponding average inference time (**Table 3**). Since the LMM part of ALToLLM-8B (FL) follows the architecture design in HiMTok, it should have the same inference time.
>
> > **Q7: What is the main factor that makes the difference between Fix-length ALTo with Fix-length tokenization compared with HimTok in Table 2 and 4?**
>
> **A:** Please kindly refer to **Q3 & 7** above.

---

> ### Comment · Reviewer_bQXT · 2025-08-04
>
> Thanks to the authors for the response.
>
> After reading the rebuttal, I still have concerns about the application scope of the adaptive length token mechanism:
>
> 1. In the task that does not require a segmentation mask (image or region understanding), how can we apply the adaptive token strategy, since no ground truth mask is available in this task, while in the paper, we need a segmentation mask to train the model?
> 2. Why are the results not better than Glamm in the Flickr 30k dataset in terms of CIDEr metrics, even with the fine-tuned models? And in this setting, what is the advantage of the proposed methods and Glamm baselines in terms of the number of tokens?

---

> > ### Author Response · Authors · 2025-08-05
> >
> > Thank you for carefully reading our response.
> >
> > >  **Q1 & Q2.2: Adaptive token strategy for image or region understanding task**
> >
> > **A:** (1) In this work, the adaptive-length tokenization is indeed applied only to mask representations, and not to regular text or RGB images. (2) However, it is important to note that we adopt a **multi-task mixed training** strategy during SFT (L192), which allows our MLLM to acquire strong segmentation capabilities while retaining its general understanding ability. The training data includes not only mask-related samples but also general image-text and pure text data (as in HiMTok[16]), enabling the model to handle diverse input-output modalities flexibly.
> > (3) The adaptive tokenization methodology is promising to be extended to other modalities like image generation in the future.
> >
> > >  **Q2.1: Why are the results not better than Glamm in the Flickr 30k dataset in terms of CIDEr metrics, even with the fine-tuned models?**
> >
> > **A:**
> > (1) We believe this result is partly due to how CIDEr works. CIDEr averages n-gram similarity, which can penalize longer, more descriptive captions by "diluting" the score, especially on datasets like Flickr30k with only five short ground-truth sentences per image (ten in NoCaps dataset). In contrast, metrics like SPICE, which evaluate semantic correctness by parsing objects, attributes, and relations, might be more suitable. Our model achieves a high SPICE score, demonstrating strong semantic understanding, even if the CIDEr score is lower.
> > (2) Our fine-tuned model learned the short-description style from a small subset of COCO data, rather than from the Flickr30k dataset.

---

> > > ### Comment · Reviewer_bQXT · 2025-08-08
> > >
> > > Thank you for your clarification. Most of my concerns are now addressed. I will carefully re-check my rate.

---

> > > > ### Author Response · Authors · 2025-08-09
> > > >
> > > > Thank you! We're glad we could address your concerns.

---

### Official Review · Reviewer_QGmw · 2025-07-02

**Clarity:** 3
**Significance:** 2
**Originality:** 3
**Rating:** 4
**Confidence:** 3

**Summary:**

This paper introduces a novel adaptive-length tokenizer called ALTo. The contribution of ALTo is its ability to dynamically determine the number of mask tokens needed to represent visual objects based on their complexity, enabling efficient and flexible mask generation for segmentation tasks. At the core of ALTo is a Token Length Predictor (TLP), which predicts the optimal token length using global and local mask features, combined with a differentiable token chunking strategy and length regularization loss. This allows the model to encode segmentation masks into variable-length token sequences—ranging from 2 to 32 tokens—rather than using a fixed length, which often leads to over- or under-representation of content.

Built upon this tokenizer, the authors develop ALToLLM, a multimodal large language model trained in four stages: (1) pretraining the mask tokenizer and de-tokenizer, (2) fine-tuning the TLP, (3) supervised fine-tuning of the MLLM with adaptive mask tokens, and (4) Group Relative Policy Optimization (GRPO) to flexibly balance segmentation accuracy (measured by IoU) and token efficiency via a composite reward. Experiments across standard benchmarks demonstrate that ALToLLM achieves state-of-the-art performance while reducing average token usage and inference time.

**Questions:**

- How exactly are the tail tokens dropped during training? It would improve clarity if the paper explained this procedure in more detail.
- The explanation of the cumulative sum might be clearer if rewritten using the form $P_i = 1 - \sum_{j < i} p_j$, to emphasize the contribution of earlier tokens to the stopping probability.

**Ethical Concerns:**

["NO or VERY MINOR ethics concerns only"]

**Final Justification:**

I have read the authors' rebuttal carefully. Thank you for the detailed and thoughtful response. The clarifications adequately addressed the concerns raised in my initial review, and I have updated my score accordingly.

**Limitations:**

yes

**Paper Formatting Concerns:**

There are no formatting concerns.

**Quality:**

3

**Strengths And Weaknesses:**

### Strengths
- **Well-designed and novel Token Length Predictor (TLP):** The training strategy for TLP is innovative and empirically validated across diverse benchmarks, demonstrating its effectiveness.
- **Careful architectural thoughtfulness:** The attention to detail in TLP’s design (e.g., differentiable chunking, attention-based gating) reflects a high level of engineering insight.
- **Strong empirical results:** The model consistently achieves state-of-the-art performance while improving token efficiency, showing practical utility of adaptive-length modeling.
&nbsp;
### Weaknesses
- **Limited scope of application:** Although the proposed adaptive tokenizer has broader potential (e.g. image tokenization), the paper evaluates it solely on mask tokenization tasks. While this choice may be justified, it would be valuable to discuss why the method is especially suitable for masks and whether it can be generalized beyond this domain.
- **Lack of baseline with fixed optimal length:** The paper does not compare against a fixed-length model set to the optimal length determined by the adaptive method—this makes it hard to isolate whether the gain comes from adaptivity or just better length calibration.

---

> ### Author Rebuttal · Authors · 2025-07-30
>
> We sincerely appreciate the reviewer’s thoughtful and constructive feedback on our work. Below, we address each point with detailed clarifications.
>
> >  **W1.1: The paper evaluates it solely on mask tokenization tasks. Why the method is especially suitable for masks?**
>
> **A:** Although the color space of masks is simple, it is non-trivial and essential to generate masks that not only ground object positions but also capture fine-grained shape details, as pointed in HiMTok [16].
> Our method is well-suited for mask generation tasks, given *the recent advances in the field* and *characteristics of the task*.
>
> (1) Recent studies like FlexTok [14], HiMTok [16], have shown that representing masks or images of varying complexity may require different numbers of tokens. For masks, simple shapes can be captured with just a few tokens, while complex ones demand more. Although HiMTok adopts a hierarchical (coarse-to-fine) token sequence, it lacks the ability to adaptively determine how many tokens are needed based on the shape complexity. Building on this insight, ALTo introduces a method that enables the model to autonomously decide the token sequence length in an adaptive manner.
> (2) The quality of generated masks can be quantitatively evaluated in a straightforward manner (e.g., IoU), and segmentation instructions typically correspond to deterministic target masks—unlike the ambiguity and diversity inherent in tasks like image generation. As the first to propose and focus on a model that autonomously adapts token sequence length, we chose to validate it on mask generation rigorously to minimize the influence of confounding factors during both training and inference, before extending to more complex domains.
>
> >  **W1.2: Whether it can be generalized beyond this domain?**
>
> **A:**  As highlighted in our paper (Introduction and references [8, 9]), both  image genration and segmentation masks can be represented within a unified tokenization framework. As demonstrated by prior works like HiMTok [16] (for masks) and FlexTok [14] (for RGB images), the coarse-to-fine token generation paradigm effectively handles both domains. Our current validation on segmentation establishes the method's effectiveness for mask generation. We fully acknowledge the potential for broader applications and plan to explore image tokenization in future work.
>
> >  **W2: Lack of baseline with fixed optimal length**
>
> **A:** In fact, **Figure 7** directly compares our adaptive-length model (ALToLLM after GRPO) against the fixed-length baseline (FL) set to the TLP-determined optimal lengths. The x-axis and y-axis show the IoU and average sequence length for each experimental setting, respectively. The blue points represent the adaptive-length models obtained with different GRPO penalty coefficients, while the red points represent the fixed-length baseline. When we compare models with the same average token length (i.e., at the same y-coordinate), the adaptive model consistently achieves higher IoU than its fixed-length counterpart, demonstrating that the performance gain stems from adaptivity itself, not merely from better length calibration. Furthermore, when comparing models with similar IoU (i.e., at the same x-coordinate), the results show that ALToLLM maintains comparable IoU while reducing token usage by more than 5 tokens per sample on average. This efficiency advantage holds consistently across all tested datasets, including zero-shot scenarios, confirming that dynamic length adaptation is a key factor underlying the superiority of our method.
>
> >  **Q1: The tail tokens' dropping approach**
>
> **A:** The procedure is carefully designed across different training stages. **Stage 1**: We randomly drop tail tokens (keeping 1-32 tokens) during training to learn hierarchical mask representation. **Stage 1.5**: We introduce the differentiable *Token Length Predictor (TLP)* which learns to optimally truncate tokens through the cumulative stopping probability. This allows the model to: (a) predict appropriate lengths based on content complexity, and (b) maintain *differentiability* for end-to-end training via our straight-through estimator. **Stage 2**: During training, we use the pretrained TLP to determine optimal sequence lengths and perform deterministic truncation. For inference, the model dynamically generates mask tokens until predicting the end token, eliminating the need for explicit token dropping.
>
> >  **Q2: Rewritten of fomulation**
>
> **A:** Thanks for this valuable and careful suggestion. We will revise the formulation to emphasize the contribution of earlier tokens to the stopping decision. We will carefully check all equations throughout the paper to ensure mathematical rigor and clarity.

---

### Official Review · Reviewer_sfeq · 2025-07-08

**Clarity:** 3
**Significance:** 2
**Originality:** 3
**Rating:** 5
**Confidence:** 4

**Summary:**

The paper modifies the HimTok approach to image segmentation in several ways, obtaining better IoU scores on multiple referring expression and open vocabulary segmentation datasets.
HimTok originally uses multimodal LLMs to perform image segmentation by expanding the LLM vocabulary size with discrete mask tokens, training the multimodal LLM to predict mask tokens, and outputting segmentation masks based on the predicted mask tokens.
The proposed modifications include (1) adaptively using a variable number of tokens to predict a segmentation mask, (2) adding a pixel encoder to provide additional information to the mask de-tokenizer, (3) using GRPO to tune the multimodal LLM, and (4) freezing and finetuning different parts of the model during different stages. (e.g. freezing everything except the LLM versus only freezing the mask tokenizer during Stage 2)

**Questions:**

In addition to the limited ablation studies mentioned above, some details are unclear.
1. What is the mask token codebook size? 1024 as in HimTok?
2. The reward function seems to suggest that the composite reward is non-zero for an incorrectly formatted sample. Does this mean that the incorrectly formatted sample is still somehow given to the mask de-tokenizer for generation?

Also, at the end of the introduction, it is mentioned that for most cases the length is 16, and some examples are given in the first page and the appendix. I'm curious if the authors have a plot of the distribution of mask token sequence length? And are there any oddities where the model uses many tokens for a seemingly simple mask, or few tokens for a seemingly complex mask?

**Ethical Concerns:**

["NO or VERY MINOR ethics concerns only"]

**Final Justification:**

The rebuttal has resolved my original concerns about the lack of ablation studies investigating the effects of each modification to the pipeline. I believe that evaluating the proposed method on mask generation is sufficient, and the additional results on image understanding and region understanding is a plus, showing potential applicability of the method to other tasks.

Hence, I have raised my score to Accept.

**Limitations:**

Yes

**Quality:**

3

**Strengths And Weaknesses:**

Strengths

The idea of allowing the model to adaptively decide how many tokens to use to represent an image is interesting.
The final pipeline brings improvements to both accuracy and efficiency (when compared against the fixed-length pipeline).
The length penalty hyperparameters for the loss function in Stage 1.5 and Stage 3  are also reasonably well studied, showing the exact trade-off between IoU and average mask token sequence length in the final models after Stage 3, as well as advantages of using 0.01 as the length penalty in Stage 1.5.


Weaknesses

My main concern is the lack of ablation studies:
Considering the high similarity to HimTok, it is crucial to investigate the effects of each modification to the pipeline. However, only the adaptive token length predictor and the pixel encoder are studied, and only to a limited degree. This leaves the reader uncertain where the gains over HimTok come from. For example, does ALToLLM still always outperform HimTok when no pixel encoders are used? What about when no GRPO is used? What if HimTok also only trained the LLM in Stage 2 training? Or vice versa, what if ALToLLM only freezes the mask tokenizer during Stage 2?

---

> ### Author Rebuttal · Authors · 2025-07-30
>
> We sincerely thank the reviewers for their thoughtful feedback and for recognizing the merits of ALToLLM, particularly its adaptive token length mechanism, improved accuracy-efficiency trade-off, and insightful analysis of length regularization.
>
> > **W1: Lack of ablation studies.**
>
> **A:** Good suggestions. To more comprehensively illustrate the key factors behind ALToLLM's performance gains over HiMTok, we conducted additional ablation studies on the *gRefCOCO val* set, although Table 2 and Figure 7 have already presented part of the relevant results.
>
> We analyze the following design choices:
> - **Pixel Encoder**: Introducing a pixel-level encoder to supply fine-grained visual features to the mask de-tokenizer.
> - **Tuning Mask De-tokenizer (MD)**: Whether the mask de-tokenizer is updated or kept frozen during Stage 2 training.
> - **Token Length Predictor (TLP)**: Predicting the number of mask tokens adaptively, enabling variable-length sequences.
> - **GRPO Training**: Applying GRPO in Stage 3 with a length penalty coefficient of $1 \times 10^{-3}$ for sequence regularization.
> - **Multi-Target-SA1B used in Pretraining**: Leveraging our dataset containing complex and multi-object masks during Stage 1 pretraining.
>
> The results are summarized below:
>
> | Method                                      | Multi-Target-SA1B | Pixel Encoder | Tuning MD | TLP  | GRPO  | cIoU  | gIoU  | Avg. Length |
> |--------------------------------------------|:-----------------:|:-------------:|:--------:|:-----:|:-----:|:-----:|:-----:|:-----------:|
> | HimTok                                     |                   |               |          |       |       | 70.4  | 72.1  |     32.0    |
> | ALToLLM                |         ✓         |               |          |       |       | 72.8  | 75.6  |     32.0    |
> | ALToLLM                |         ✓         |       ✓       |          |       |       | 74.8  | 77.6  |     32.0    |
> | ALToLLM                |         ✓         |       ✓       |    ✓     |       |       | 74.7  | 77.6  |     32.0    |
> | ALToLLM                |         ✓         |       ✓       |          |   ✓   |       | 75.4  | 78.0  |     17.5    |
> | ALToLLM                |         ✓         |       ✓       |          |   ✓   |   ✓   | **75.5**  | **78.1**  |     **15.7**    |
>
> > *W1.1: Where the gains over HimTok come from.*
>
> **A:** The ablation study reveals that the performance improvements of ALToLLM over HimTok primarily stem from two key components:
> (1) **Multi-Target-SA1B Pretraining** boosts cIoU from 70.4 to 72.8, significantly enhancing the model’s ability to handle scenes with multiple and complex objects.
> (2) The **Pixel Encoder** further increases cIoU by 2.0 points (to 74.8), thanks to the fine-grained visual feature that enables more accurate mask boundary reconstruction.
>
> Additionally, the **TLP** improves token efficiency—reducing average output length by around 50%—while still increasing cIoU to 75.4, indicating that adaptive token generation reduces redundancy without sacrificing quality.
>
> > *W1.2: Effect of GRPO*
>
> **A:** We acknowledge that GRPO contributes only marginally to absolute performance metrics (e.g., +0.1 cIoU). However, as discussed in **L170**, its primary value lies in **controllability**: GRPO offers a flexible mechanism to balance mask quality and token efficiency.  As shown in **Figure 7**, increasing the length penalty coefficient in GRPO training improves token efficiency at varying degrees of cost to mask quality. This tunability allows us to adapt the model to different application needs—prioritizing accuracy in high-fidelity scenarios or efficiency in bandwidth-constrained environments. Moreover, as demonstrated in **Figure 6**, this adaptation can be achieved within just a few hundred training steps, highlighting the efficiency and practicality of GRPO training in dynamically shaping model behavior.
>
> > *W1.3: Difference from HimTok in Stage 2 training*
>
> **A:** In Stage 2, unlike HimTok—which jointly fine-tunes both the mask de-tokenizer (MD) and the multimodal large language model (MLLM)—we only train the MLLM while keeping the MD frozen.  Our ablation shows this design achieves comparable performance, as MD has been already well-learned during Stage 1 and capable of accurately generating diverse masks. Notably, we observed that the public implementation of HimTok omits the original LayerNorm layer when it is integrated into LMM, which is inconsistent and necessitate the tuning of MD in Stage 2. In contrast, our architecture is consistent before and after integrating ALTo into LMM, which eliminates the need to update the MD in Stage 2.
>
> >  **Q1: Mask token codebook size clarification**
>
> **A:** The mask token codebook size is 1024, consistent with HimTok for fair comparison.
>
> >  **Q2: Rewards about incorrectly formatted sample**
>
> **A:** In our mask de-tokenizer step, any responses that do not conform to the correct format will raise an exception, and its corresponding IoU reward is set to 0. As a result, incorrectly formatted samples always receive lower composite rewards compared to correctly formatted ones. We have empirically observed the desired behavior with this design: during GRPO training, the average format reward consistently converges to 1 within approximately 20 steps, and thereafter the model rarely produces samples with formatting errors.
>
> >  **Q3: Provide a plot of the distribution of mask token sequence length**
>
> **A:** (1) In **Figure 8**, we present the distribution of mask token sequence lengths in Multi-Target-SA1B under different penalty coefficients during Stage 1.5. When the coefficient is set to 0.01, the token lengths are primarily concentrated between 20 and 30, which may be attributed to the more complicated masks in Multi-Target-SA1B. (2) To see the actual distribution on real benchmarks, we plotted the token length distribution of ALToLLM (SFT) on the *gRefCOCO val* set, where most samples fall within the 10–20 range. Due to rebuttal constraints, we present the distribution for lengths 10–20 in tabular form instead of images.
>
> | Token Length | 10  | 11  | 12  | 13  | 14  | 15  | 16  | 17  | 18  | 19  | 20  |
> |--------------|-----|-----|-----|-----|-----|-----|-----|-----|-----|-----|-----|
> | Sample Number| 190 | 286 | 436 | 551 | 378 | 291 | 498 | 526 | 325 | 424 | 432 |

---

> > ### Comment · Reviewer_sfeq · 2025-08-07
> >
> > Thank you for the detailed new ablations. I am better convinced that each of the proposed modifications contribute some improvement. However, I have some further questions:
> >
> > **W1**
> > When I first read the paper, I thought that the FL and AL models in Table 2 were comparing a fixed length model with a fixed length model + length predictor + GRPO training.
> >
> > According to Row 5 of the new table, which is the same as the last row in Table 2, I assume that the AL model in Table 2 (as well as the AL models in Table 3~ Table 7) i is not trained with GRPO? This was not made clear in the paper. So is it that only the results in Figure 6 to Figure 8 that involve GRPO?
> >
> > Furthermore, can you clarify why there is a difference between the average mask token length for Row 4 and Row 5 of the new ablations (without or with GRPO)? My original understanding was that the Stage 3 GRPO training only finetunes the MLLM parameters, without changing the MT, TLP, or MD. If so, since response length is determined by the TLP, shouldn't the sampled responses before and after GRPO training have the same average lengths?
> >
> > Lastly, you mention that HimTok may require MD tuning due to a missing LayerNorm. Are you referring to a LayerNorm somewhere in the mask tokenizer? Or is it somewhere else?
> >
> > **Q2**
> >
> > I see. I think it is worth adding one line mentioning that IoU is 0 for incorrectly formatted samples. So the incorrect formatted samples have strictly negative rewards from the length penalty term?
> >
> > **Q3**
> >
> > Thank you. It would be interesting to see if there were any edge cases where the model uses surprisingly many or surprisingly few tokens, but this is a minor thing.

---

> > > ### Author Response · Authors · 2025-08-07
> > >
> > > Thank you for carefully reading our response.
> > >
> > > >**W1.1: Confusion on SFT and GRPO results**
> > >
> > > We apologize for the confusion. Your understanding is correct. The results in Tables 2-7 are from the SFT stage. Results involving GRPO are presented in Figures 6-7.
> > >
> > > >**W1.2: Difference in average mask token length between SFT and GRPO**
> > >
> > > For MLLM, the TLP's role is confined to the SFT stage, where it provides ground-truth (various-length mask token sequences) supervision. This process teaches the MLLM to dynamically generate mask tokens until it predicts an end token, i.e., <ALTo_End>.
> > > You are correct that GRPO only finetunes the MLLM, leaving the MT, TLP, and MD unchanged. By applying varying degrees of length penalty (\alpha in Eq.2), different token lengths are preferred respectively. MLLM learns to output the <ALTo_End> token after adaptive number of mask tokens are generated.
> > >
> > > >**W1.3: Missed LayerNorm in HiMTok**
> > >
> > > Yes, your understanding is correct. We are referring to the LayerNorm located just before the vector quantization step inside the mask tokenizer.
> > >
> > > >**Q2: Rewards for incorrect formatted samples**
> > >
> > > Thank you for the suggestion. We will clarify in the paper that the IoU is 0 for incorrectly formatted samples. Consequently, these samples receive a strictly negative reward from the length penalty term.
> > >
> > > >**Q3: Edge cases**
> > >
> > > That is a great point. We would be happy to show such edge cases. However, due to the format constraints of the rebuttal, we can only present these findings in the table below.
> > >
> > > | Object Description | Number of Mask Tokens | Number of Objects |
> > > |--------------------------|-----------------------|-------------------|
> > > | the woman on the left | 7 | 1 |
> > > | + the woman on the right | 18 | 2 |
> > > | + suitcase | 20 | 3 |
> > > | + hat | 22 | 4 |
> > > | + blanket | 25 | 5 |
> > > | + flower on the table | 30 | 6 |
> > >
> > > The table shows that as more objects are described, more mask tokens are consumed. For instance, describing only 'the woman on the left' is an edge case using few tokens, while describing all objects is an edge case using many.

---

> > > > ### Author Response · Authors · 2025-08-09
> > > >
> > > > Dear Reviewer,
> > > >
> > > > I hope this message finds you well. As the discussion period is nearing its end with less than one day remaining, I wanted to ensure we have addressed all your concerns
> > > > satisfactorily. If there are any additional points or feedback you'd like us to consider, please let us know. Your insights are invaluable to us, and we're eager to address any remaining issues to improve our work.
> > > >
> > > > Thank you for your time and effort in reviewing our paper.

---

> > > > > ### Comment · Reviewer_sfeq · 2025-08-09
> > > > >
> > > > > **W1.2**: I see. I originally thought that the TLP will also be used during inference to force the output length of the MLLM to be exactly as the TLP predicts.
> > > > >
> > > > > Thank you for the clarifications, I have no further questions.
> > > > > The rebuttal has resolved my concerns, and I will raise my rating to reflect this.
> > > > >
> > > > > I would strongly suggest the authors to emphasize that results in Tables 2-7 are from the SFT stage, as well as incorporating the new ablation results in W1 and relevant discussions in the final revision.

---

### Comment · Area_Chair_XuQ6 · 2025-08-05
**please do the post-rebuttal action items**

Dear reviewers,

For the reviewers who have not yet read the authors' rebuttal and the other reviews, **please do so now**. Per the NeurIPS guidelines, the reviewers must comment if the authors' response did or did not address their concern, and **the reviewers cannot enter the "mandatory acknowledgment" before they have sent a comment on whether the authors' responses did/didn't address their concern.**

**Please read the reviews and rebuttal, let the authors know, and submit your final score accordingly**. The NeurIPS chairs specifically directed the AC to ask you not to wait till the last moment for this in order not to defeat the purpose of the reviewer-author discussion period.

Thank you for your service!

-Area Chair

---

### Note · Authors · 2025-08-12

We thank all reviewers for their thorough assessment of our work. Our approach is well recognized as **interesting** (R1,R3), **intuitive** (R3), **reasonable** (R4), **innovative** (R2), **effective** (R2,R3) and **practical** (R2), with several key advantages:

1. **Adaptive Token Length Mechanism**: As reviewers noted, our approach enables the model to **automatically decide the exact token length for mask generation**, based on complexity.

2. **Balanced Performance-Efficiency Trade-off**: Our approach successfully **balances mask quality and computational efficiency** through GRPO, achieving optimized token usage while maintaining the IoU score.

3. **Comprehensive Validation**: The effectiveness of our approach is validated across diverse benchmarks, demonstrating **state-of-the-art performance while improving token efficiency** compared to fixed-length approaches.

We have addressed the main concerns as follows:

1. **Additional ablation studies**: We have provided **comprehensive ablation experiments as requested, which helps clarify the contribution of each component in ALTo.

2. **Performance-Efficiency Trade-off via GRPO**: Figures 6-7 in our paper demonstrate how GRPO effectively balances segmentation quality and computational efficiency.

3. **Generalizability beyond mask generation**:  Our method also demonstrates competitive performance on other tasks like reasoning segmentation and region understanding. Additionally, we have explained how our adaptive token length mechanism can be extended to other autoregressive tasks beyond segmentation, such as image generation.

4. **Clarification of technical details**: We have addressed all ambiguities pointed out by reviewers, providing clearer explanations of our methodology and implementation details.

---

### Decision · Program_Chairs · 2025-09-17

**Decision:**

Accept (poster)

**Comment:**

The submission address tokenization in image generation and multimodal frameworks, specifically the fact that the tokens often (but not always) conform to a rigid and fixed length structure, irrespective of the content. They propose a tokenizer that utilizes an adaptive length, employing a token length predictor, a length regularizer, and a token chunking technique.

This is an important problem acknowledged by many in the community, with numerous parallel efforts underway. Despite that, all reviewers recognize that the submission has novel merits and vote for acceptance, though mostly with a borderline stance. The AC agrees and recommends acceptance.